# Multiscale structural control of thiostannate chalcogels with two-dimensional crystalline constituents

Thanh Duy Cam Ha[1,8], Heehyeon Lee[2,3,8], Yeo Kyung Kang[1], Kyunghan Ahn[1], Hyeong Min Jin[4], In Chung[5,6], Byungman Kang[7], Youngtak Oh[3] ✉ & Myung-Gil Kim[1] ✉

Chalcogenide aerogels (chalcogels) are amorphous structures widely known for their lack of localized structural control. This study, however, demonstrates a precise multiscale structural control through a thiostannate motif ($[Sn_2S_6]^{4-}$)-transformation-induced self-assembly, yielding Na-Mn-Sn-S, Na-Mg-Sn-S, and Na-Sn(II)-Sn(IV)-S aerogels. The aerogels exhibited $[Sn_2S_6]^{4-}$:$Mn^{2+}$ stoichiometric-variation-induced-control of average specific surface areas (95–226 $m^2 \, g^{-1}$), thiostannate coordination networks (octahedral to tetrahedral), phase crystallinity (crystalline to amorphous), and hierarchical porous structures (micropore-intensive to mixed-pore state). In addition, these chalcogels successfully adopted the structural motifs and ion-exchange principles of two-dimensional layered metal sulfides ($K_{2x}Mn_xSn_{3-x}S_6$, KMS-1), featuring a layer-by-layer stacking structure and effective radionuclide ($Cs^+$, $Sr^{2+}$)-control functionality. The thiostannate cluster-based gelation principle can be extended to afford Na-Mg-Sn-S and Na-Sn(II)-Sn(IV)-S chalcogels with the same structural features as the Na-Mn-Sn-S chalcogels (NMSCs). The study of NMSCs and their chalcogel family proves that the self-assembly principle of two-dimensional chalcogenide clusters can be used to design unique chalcogels with unprecedented structural hierarchy.

An aerogel is a gel-derived porous material with ultra-low density (0.0011–0.5 $g \, cm^{-3}$), well-developed porous channels with sizable specific surface areas (32–3000 $m^2 \, g^{-1}$) and pore volumes (0.2–3.38 $cm^3 \, g^{-1}$), a random orientation and porosity ranging from the nano- to macro-level[1–5]. Aerogels have been synthesized using oxides, carbon, metals, biopolymers, resins, and other constituents, yielding diverse arrays of physiochemical, structural, and functional properties. Metal-oxide aerogels—the most prominent variety—have applications

in catalysis[6], supercapacitors[7], batteries[8], environmental remediation[9], photocatalysis[10], sensors[11], and thermal insulators[12]. The excellent physicochemical stability and convenient sintering of typical oxide aerogels enable the development of a well-defined structure and uniform surface with desirable functionalities. However, the robustness of the oxide surface often limits its functionality in applications requiring dynamic and selective electronic, chemical, and photonic molecular interactions. In these areas, chalcogenide aerogels (chalcogels)

[1]School of Advanced Materials Science & Engineering, Sungkyunkwan University, Suwon 16491, Republic of Korea. [2]Department of Materials Science and Engineering, Korea University, Seoul 02841, Republic of Korea. [3]Center for Sustainable Environment Research, Korea Institute of Science and Technology, Seoul 02792, Republic of Korea. [4]Department of Organic Materials Engineering, Chungnam National University, Daejeon 34134, Republic of Korea. [5]School of Chemical and Biological Engineering, and Institute of Chemical Processes, Seoul National University, Seoul, Republic of Korea. [6]Center for Correlated Electron Systems, Institute for Basic Science (IBS), Seoul, Republic of Korea. [7]Nuclear Chemistry Research Division, Korea Atomic Energy Research Institute, Daejeon 34057, Republic of Korea. [8]These authors contributed equally: Thanh Duy Cam Ha, Heehyeon Lee. ✉e-mail: ytoh@kist.re.kr; myunggil@skku.edu

containing chalcogen (S, Se, or Te) and their counterbalancing inorganic components have a significant potential.

Chalcogels are unique inorganic porous structures containing chalcogenide clusters with various primary coordination; tetrahedral ($[MQ_4]^{4-}$ and $[M_2Q_6]^{4-}$, M = Mo, W, Sn, Ge; Q = S, Se)[1,4,13], adamantine clusters ($[M_4Q_{10}]^{4-}$, M = Sn, Ge/Q = S, Se)[14], ditopic linear polychalcogenide anions ($S_x^{2-}$; $x = 3–6$)[15,16], trigonal pyramidal anions ($[XS_3]^{3-}$, X = As, Sb)[17], triangular molybdenum sulfide clusters ($[Mo_3S_{13}]^{2-}$)[18], multinary chalcogenide clusters[17,19], and chalcophosphates ($[P_2S_6]^{2-}$)[20] are used as the main constituents of ionic self-assembly. Conventionally, the sol−gel metathesis reaction between the chalcogenide cluster anions and linker counterions develops a three-dimensional random porous network with a highly polarizable and Lewis-basic porous surface. Consequently, these form the centers for selective soft-Lewis-acid cation attraction, enabling the application of chalcogels in water treatment[21,22], catalysis[23,24], gas adsorption[25], and radionuclide control[26]. However, uncontrolled and spontaneous metathesis reaction generally yields disordered domains without adequate control over the macroscopic alignment of the molecular integrity. Overall, a precise control of the porous structure is limited by the finite choices of chalcogenide constituents and the absence of structural directing agents during metathesis.

The three-dimensional gel structure of most metathesis-based chalcogels can be transformed into dried form (aerogel), while the original chalcogenide cluster and transition metal-linker constituents maintain their chemical integrity. In general, the transition metal linker is a prerequisite for most chalcogel development. However, a recent chalcogel study by Kang et al. reported the first case of a transition metal-linker-free chalcogel synthesis[13], in which the thiostannate cluster underwent a tin center-coordination transformation, triggering self-gelation. This study suggests that chalcogel chemistry is no longer bound to limited combinations of few stable transition metal linkers and fixed chalcogenide cluster structural motifs, enabling flexible gel architecture for desirable functionalities. Based on this confirmed speculation, a three-dimensional porous architecture with the unique molecular dynamics of the two-dimensional chalcogenide structure, featuring both an exotic functionality of two-dimensional surface and a highly active three-dimensional porous network, is possible. For instance, the effective ion-exchange principle of the well-known two-dimensional layered metal sulfides $K_{2x}Mn_xSn_{3-x}S_6$ (KMS-1) can be adopted into a three-dimensional platform if the chalcogel with such a layer-structure motif can be synthesized. Graphene oxide chemistry, wherein numerous porous graphene oxides have been explored for various functionalities, also demonstrates this structural architecture design concept (two-dimensional functionality in a three-dimensional structure) well[27,28].

Inspired by coordination-transformation-induced chalcogel formation and the three-dimensional graphene oxide porous structure architecture, we envisioned the possibility of designing a three-dimensional chalcogel network containing two-dimensional local structural motifs, to afford a multiscale structural control of chalcogels with moderate-range atomic ordering, well-defined mesoporous channels, and long-range macroscale porosity. Here, we report the successful synthesis of layered Na-Mn-Sn-S chalcogels (NMSCs) using thiostannate clusters and manganese ions. To the best of our knowledge, this is the first chalcogel system containing a crystalline layer structural motif with a well-defined three-dimensional porous network. As an alternative to the classical metathesis reaction, this NMSC involved a thiostannate-coordination transformation from tetrahedral to octahedral with a spontaneous ligand-switching (metathesis) interaction between the thiostannate cluster and manganese linker. The manganese linker triggered a crystalline layer-structure assembly, while the sodium sites acted as ion-exchange centers to afford a hard-to-intermediate cation-capture functionality. Using stoichiometric variations of the $[Sn_2S_6]^{4-}:Mn^{2+}$ ratio, the crystallinity of the two-

dimensional structural motif in the NMSC was modified from amorphous to layered crystalline by kinetically controlling the $Mn^{2+}$–thiostannate metathesis and mode transformation of the tin center in the thiostannate. The resulting crystalline NMSC feature an ion-exchangeable alkali metal site, rapid mass transportability through the porous network, and multiscale porous channels between chalcogel layers ranging from meso- to macro porosity. These structural features of NMSC allowed effective radionuclide ($Cs^+$ and $Sr^{2+}$) remediation functionality. Furthermore, a kinetically controlled coordination-transformation principle was implemented for a hierarchical aerogel synthesis of layered chalcogenides, as demonstrated by the Na-Mg-Sn-S and Na-Sn(II)-Sn(IV)-S aerogels. These chalcogels demonstrates the diversity and expendability of thiostannate-cluster chemistry, which have a significant potential for the design of unique porous materials with desirable functionalities from well-defined two-dimensional constituents.

## Results and discussion
### Synthesis, characterization, and coordination chemistry of the NMSCs
The randomly distributed porous structure of previously reported chalcogels was transformed by the self-assembly reaction with the three-dimensional chalcogenide constituents. However, unlike the conventional metal-linker chalcogels, $Na_{1.71}Mn_{1.09}Sn_2S_{4.92}$ (NMSC-1, $[Sn_2S_6]^{4-}:Mn^{2+}$ ratio = 1:0.5) structural motifs were developed within their gel framework due to complete gelation of $Na^+$, $Mn^{2+}$, and $[Sn_2S_6]^{4-}$ via the metathesis reaction and a simultaneous coordination transformation (Fig. 1a). NMSCs designated as NMSC-2 and NMSC-3 were synthesized by changing the $[Sn_2S_6]^{4-}:Mn^{2+}$ ratio from 1:0.5 to 1:1 and 1:2, respectively.

As shown in Fig. 1a, a transparent yellow precursor solution containing $Na^+$, $Mn^{2+}$, and $[Sn_2S_6]^{4-}$ turned to a light-amber turbid solution over 5 h, and subsequently to a darker wet gel whose viscosity increased until a mechanically integrated monolithic gel was formed in 7 d. With subsequent solvent-exchange and critical point drying (CPD) processes, a monolithic and homogeneous light-colored aerogel was produced, retaining the volume of its wet-gel state (Fig. 1a and Supplementary Figs. 1, 2). The uniform distribution of Na, Mn, Sn, and S in the resulting aerogel proved a successful assembly of the inorganic constituents, as indicated by the energy dispersive spectrometry (EDS) mapping result (Fig. 1b). The coexistence of a thin-layer morphology and wrinkle texture (like a crumpled paper ball) in the scanning electron miscrocopy (SEM) image of the synthesized NMSC suggested a two-dimensional layer-structure assembly of the constituents and a secondary self-crosslinking reaction to yield a three-dimensional geometry. The unique geometric morphology of the NMSCs, unlike the sponge-like amorphous porous morphologies of previously reported chalcogels, indicated unprecedented structural transformation during the sol−gel reaction.

To understand the unique structural transformation of NMSCs and define their structural identity, NMSC-1 aerogel was analyzed using powder X-ray diffraction (PXRD), Fourier transform infrared spectroscopy (FT-IR), and X-ray photoelectron spectroscopy (XPS) (Fig. 1c–f). The PXRD analysis of the NMSCs provided crucial information regarding the structural identity of their constituents (Fig. 1c and Supplementary Fig. 3). Unlike conventional amorphous chalcogels with no apparent PXRD peaks, NMSC-1 revealed notable phase information via PXRD, with prominent diffraction peaks at 5°, 9.8°, 19.8°, 28.7°, 33° (with a shoulder peak at 36–42°), and 50.2°. The main peaks of NMSC-1 resemble the Bragg diffraction peaks (10.1°, 20.3°, 28.7°, 33.0°, and 50.5°) of $K_{2x}Mn_xSn_{3-x}S_6$ (KMS-1), a well-known layered metal sulfide (Supplementary Fig. 4)[29]. Furthermore, the crystal structure pattern of NMSC-1 is well confirmed by the PXRD patterns of NMS and KMS-1 bulk materials (Supplementary Fig. 5). The chemical composition and structural pattern of NMSC resemble the KMS−1 structure,

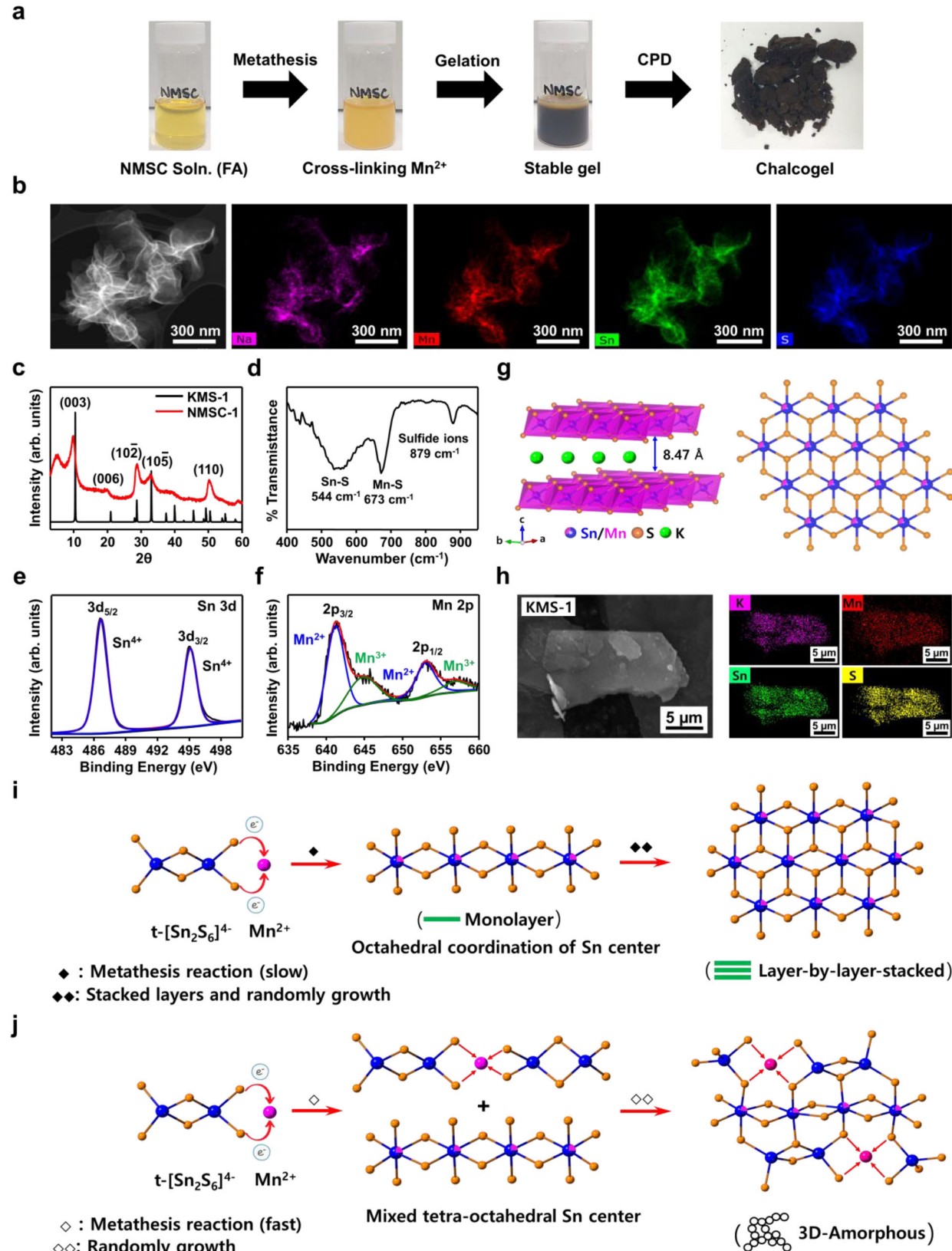

**Fig. 1 | Synthesis, characterization, and hypothesized mechanism of coordination (Tetrahedral-Octahedral)-structure (Layer–Amorphous) switching of sodium manganese sulfide chalcogel (NMSC) with a layered structure similar to that of metal sulfide (KMS-1) materials. a** NMSC synthesis. **b** EDS mapping of layered NMSC-1, including Na, Mn, Sn, and S. **c**–**f** Structural analyses of NMSC-1 using PXRD, FT-IR, and XPS analyses of NMSC-1 for Sn 3*d* and Mn 2*p*, respectively. **g**, **h** Scheme and EDS mapping of layered KMS-1 structure, including K, Mn, Sn, and S. **i**, **j** Hypothesis mechanism of coordination (Tetrahedral-Octahedral) and structure (Layer–Amorphous) transformation in NMSC. Source data are provided as a Source Data file.

wherein manganese, tin, and sulfide were assembled into a layered structure with an alkali metal ($Na^+$ for NMSC and $K^+$ for KMS-1) in the interlayer position. The prominent peaks of NMSC-1 at 9.8° and 19.8° 2θ are the representative (003) and (006) reflections of c-axis lengths in the $R\bar{3}m$ space group (Supplementary Fig. 6)[30], corresponding to an interlayer distance of 9.02 Å and a c-axis length of 27.06 Å. The left-shifted XRD peaks (9.8°, 19.8° vs. 10.1°, 20.3°) indicate an increased interlayer distance (9.02 Å) and c-axis length (27.06 Å) in NMSC-1 compared to KMS-1. In addition, the 50.2° 2θ peak corresponds to the a-axis double d-spacing value for the (110) reflection, which had a Sn−Sn distance of 3.63 Å (NMSC-1) that was proportional to the Sn/Mn−S distance[29]. As opposed to the 3.68–3.69 Å of the KMS-1 structure, the 3.63 Å of NMSC-1 indicates a potentially mixed-valence state of $Mn^{2+}$ and $Mn^{3+}$ in such a reduction in Sn/Mn−S bonding distance. The other two peaks at 28.7° and 33° 2θ are the (10$\bar{2}$) and (10$\bar{5}$) reflections of the $R\bar{3}m$ space group. The absence of a low angle peak around 5° 2θ (17.66 Å) in the KMS−1 structure indicates that the supercell phenomenon, which results in a crystal-structural parameter (c-axis) twice as high as the NMSC-layer distance, leads to a poorly stacked layer-structure assembly during the sol–gel reaction of NMSC-1[31,32].

The single-crystal XRD of KMS-1 was compared with NMSC-1 to understand the distinctive nature of crystallinity. The single-crystal XRD of KMS-1 revealed a perfectly ordered ABC stacking, indicating a defect-free uniformly layered distance. The polycrystalline NMSC-layer structure, in contrast, is expected to have a moderate-range periodic arrangement with defective domains such as lattice disorder, dislocations, grain boundaries, and interlayer defects (stacking faults). As the PXRD results indicate isostructural NMSC-1 and KMS-1 with slight unit-cell deviations (due to lattice compression/expansion), the core coordination of NMSC-1 is expected to resemble that of the KMS-1 structure with an octahedral Mn/Sn center. This means that the original tetrahedral coordination of the thiostannate cluster transforms into an octahedral coordination during the sol–gel self-assembly reaction.

The small-angle (SAXS) X-ray scattering provide further information regarding the electron-density structure in materials at small scattering angles, which helps in understanding the size, shape, and pore size (SAXS) in partially ordered materials like NMSCs (Supplementary Fig. 7). The SAXS profile of NMSC−1 contains a shoulder-shaped scattering peak at 0.3 Å$^{-1}$ (Level 1) with a monotonic decay at lower q. This hump corresponds to a nanometer-sized scattering object attributed to the microstructure of the NMSC matrix. The scattering in the lower q region near 0.01 Å$^{-1}$ originates from larger structures such as meso-macropores and particles in the range 30–50 nm. From the unified fit for Level-1 scattering, a radius of gyration ($R_g$) of 5.3 Å was obtained for the NMSC matrix unit. $R_g$ can be attributed to the NMSC stacking cluster units, as micropores were not observed in the NMSC. Assuming a spherical shape, the average cluster diameter of NMSC−1 was 1.4 nm (i.e., d = 2·(5/3)$^{1/2}$·$R_g$), corresponding to multilayer stacks of octahedral coordinated manganese-thiostannate-cluster (o-MTC) units (1.7–1.8 nm, Supplementary Fig. 5a–b). Scattering from the larger meso-structure (near 0.01 Å$^{-1}$, Level-2) of NMSC-1 indicates an intra-particle meso-porosity ($R_g$ = 17.8 nm, d = 46.9 nm) similar to the porosity of hierarchically mesoporous silica (Supplementary Fig. 8c–e)[33].

To understand the chemical-bonding nature and functional groups of NMSC-1, FT-IR and Raman spectroscopy were used. The FT-IR peaks at 544 (Sn−S)[34], 673 (Mn−S)[35], and 879 cm$^{-1}$ indicate resonance interactions between the vibrational modes of the sulfide ions (Fig. 1d)[35]. Similar fingerprint spectra were also obtained for KMS-1. Moreover, the Raman spectrum of $Na_4Sn_2S_6$ demonstrated the structure of [$Sn_2S_6$] cluster (Supplementary Fig. 9a). The peak at 385 cm$^{-1}$ is responsible for the resonance of symmetric Sn-S$_{terminal}$ stretching mode, and the tetrahedral Sn-S vibration in the bridging site presents at 346 cm$^{-1}$. The peak values in a range of between 346 and

385 cm$^{-1}$ are assigned to the isolated [$Sn_2S_6$]$^{4-}$ anions. The Sn-S$_{bridge}$ vibration occurs at 328 cm$^{-1}$. The $Sn_2S_2$ ring vibration is located at 287 cm$^{-1}$. The peaks below 200 cm$^{-1}$ are responsible for deformation vibration of $SnS_2$ wagging and twisting modes[36]. However, after coordinated with $Mn^{2+}$, NMSC-1 displays prominent vibrational peaks in the 271, 314, and 356 cm$^{-1}$ in Raman spectroscopy (Supplementary Fig. 9a), indicating $Sn_2S_2$ ring vibrations and Sn-S vibrations in the octahedral tin cluster[13,37]. The disappearance of Sn-S$_{terminal}$ vibration and retention of the $Sn_2S_2$ ring vibration indicate self-assembly between the terminal sulfur of [$Sn_2S_6$]$^{4-}$ cluster and $Mn^{2+}$ linker. In the NMSC-2 and NMSC-3, a peak at 348 cm$^{-1}$ presents tetrahedral Sn-S vibration in bridging site at marginally higher energy, corresponding to the presence of a tetrahedral tin center (Supplementary Fig. 9b)[38]. In addition, the peaks at 218 and 472 cm$^{-1}$ suggest Mn−S vibrations and sodium−sulfide interactions (Supplementary Fig. 9b)[39]. This indirectly proves the tetrahedral-to-octahedral transformation of the thiostannate cluster during the sol−gel self-assembly reaction.

XPS analysis of NMSC-1 provides the oxidation states of its crucial constituents and the feasible coordination modes and electron-density environments of the surrounding ligands and cationic centers. Considering the typical $3d_{5/2}$ binding energies of $Sn^{4+}$ (486.6–486.4 eV) compounds[29], the Sn 3d peaks at 486.7 for $3d_{5/2}$ and 495.0 eV for $3d_{3/2}$ (Fig. 1e) suggest a homogenous Sn(IV) oxidation state within the NMSC network. Unlike the tin state, the manganese center of NMSC contains a mixed-valence state of $Mn^{2+}$ and $Mn^{3+}$ with a 1.9:1 area ratio, indicating a dominant $Mn^{2+}$ oxidation state (Fig. 1f). The two peaks at 641.18 and 652.98 eV, with a difference of 11.8 eV, correspond to $Mn^{2+}$, with spin-orbit splitting of $2p_{3/2}$ and $2p_{1/2}$, respectively, corresponding to an area ratio of 2:1[40]. The peaks at 644.68 and 657.08 eV, with a 12.4 eV difference, can be designated to $Mn^{3+}$, with spin-orbit splitting of $2p_{3/2}$ and $2p_{1/2}$, respectively[41]. Moreover, considering that the typical splitting-energy separations ($\Delta E$) of Mn 3 s are 5.8–6.2 eV for Mn(II) and 5.2–5.5 eV for Mn(III), NMSC-1 exhibits a mixed-oxidation state of $Mn^{2+}$ and $Mn^{3+}$, with a $\Delta E$ of 5.6–5.7 eV for Mn 3 s (Supplementary Fig. 10)[42,43]. Based on the study of ion-exchange KMS-1, the mixed-valence state of $Mn^{2+}$ and $Mn^{3+}$ existed stably in crystalline structure with a larger interlayer distance investigated in ambient air or $N_2$ gas[29]. Therefore, the solution phase synthesis of NMSC with similar layer distance could result in the mixed valance state of $Mn^{2+}$ and $Mn^{3+}$ as a thermodynamically stable phase for NMSC-1. The original layer-structured KMS-1 possessed octahedral $Mn^{2+}$ as the coordination center, whereas NMSC possesses manganese in a mixed divalent and trivalent state as the center. This suggests that the manganese center of NMSC plays a pivotal role as a metastable center to afford disordered gel domains during the sol−gel reaction process. In addition, the peaks in the 160–165 eV region correspond to the 2p doublet binding energy of sulfide. A broad peak area covers the splitting of $2p_{3/2}$ (161.38 eV) and $2p_{1/2}$ (162.58 eV) by a difference of 1.16 eV, with a corresponding area ratio of 2:1 (Supplementary Fig. 11). These peaks represent Sn−S and Mn−S bonding, which overlap with the typical meta-sulfur bonding. The absence of elemental S and sulfate ([$SO_4$]$^{2-}$) indicates minimal byproduct formation and a successful homogenous gelation process of the NMSC. Supplementary Figures 10–12 show the XPS data of NMSC-2 and NMSC-3 which are similar element oxidation states to NMSC-1.

To confirm the two-dimensional local structural motif of NMSC, KMS-1, the mother structure of NMSCs, is synthesized and characterized through SEM, and EDS analysis. (Fig. 1g, h). The typical metal layer sulfides ($KMSn_2S_6$) families in the space group of $R\bar{3}m$ includes KMS-1 (M = $Mn^{2+}$)[29], KMS-2 (M = $Mg^{2+}$)[44], and KMS-5 (M = $In^{3+}$)[45]. The layer is constructed by (M/Sn)$S_6$ octahedral coordination with M and Sn atoms occupying the same position. These structures are generally obtained by high temperature and pressure hydrothermal synthesis. We anticipate that NMSCs develop the structural motif of KMS-1 with the $Na^+$ replacement at the $K^+$ position during the spontaneous gelation.

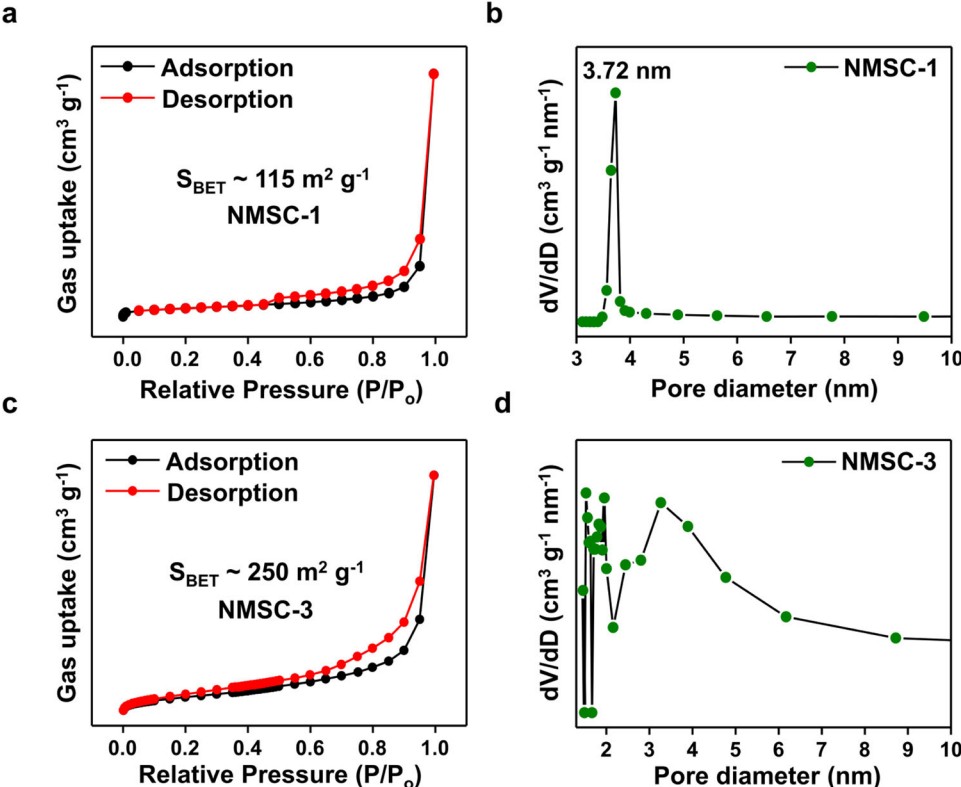

**Fig. 2 | Pore characteristic control in NMSC via stoichiometric ratio. a–d** Structural variations with specific surface areas and pore-size distributions of **a, b** layered NMSC-1 and **c, d** amorphous NMSC-3. Source data are provided as a Source Data file.

The tin coordination transformation indicates the usefulness of the stoichiometric variation of the NMSC constituents (thiostannate cluster, sodium, and manganese) for controlling the macroscopic ordering degree. All three NMSCs present a common metathesis reaction with a donor-acceptor bond between $Mn^{2+}$ and tetrahedral thiostannate cluster (t-$[Sn_2S_6]^{4-}$). If slow gelation allows thermodynamically stable phase formation, as in the case of NMSC-1 formation, the manganese center favors a densely packed octahedral coordination and causes a shift in the core coordination of the $Sn^{4+}$ center from tetrahedral to octahedral. Resulting octahedral coordinated manganese-thiostannate cluster triggers layer-by-layer stacking aggregation to form the layered NMSC-1 structure, as shown in Fig. 1i. However, increasing the molar ratio of $[Sn_2S_6]$:Mn to 1:1 (NMSC-2) or 1:2 (NMSC-3) stimulates the gelation process, leading to a mixed tetrahedral-octahedral coordination formation of the $Sn^{4+}$ center. The abundant stoichiometric metal linking centers ($Mn^{2+}$) facilitated a rapid assembly of mixed $Sn^{4+}$ coordination. As a result, the mixed coordination of $Sn^{4+}$ center creates a three-dimensional amorphous macrostructure with short-range ordering and random orientation. This random self-crosslinking behavior becomes more dominant as the stoichiometric $Mn^{2+}$ ratio increase, as in the case of NMSC-3 (Fig. 1j). Thus, we speculate that the stoichiometric-$Mn^{2+}$-variations play a crucial role in determining the coordination mode of NMSC and its corresponding porous structure.

**Pore analysis in NMSC**

To verify a successful stoichiometric control in the NMSCs, an elemental analysis of the targeted stoichiometric elements (Na, Mn, Sn, and S) was conducted using X-ray fluorescence (XRF) (Supplementary Table 1). The stoichiometric ratios of sodium and manganese to the Sn center were inversely proportional. With increasing manganese content, the sodium content decreased because of the cationic charge balance in the chalcogel structure, indicating a successful

inclusion of the manganese linkers in the Sn-S network. The reduced Sn-to-S ratio in the XRF analysis of NMSC-3 indicates that an increase in the number of metastable manganese centers could result in the sporadic inclusion of other counterions, such as acetate. This matches the FT-IR results for the NMSCs (Supplementary Fig. 13), where the inclusion of an acetate ligand in the NMSC matrix becomes more apparent with an increase in the Mn content in $[Sn_2S_6]$:Mn[46].

Due to the different structural development environments, NMSC-1, NMSC-2, and NMSC-3 show distinct porous structures, analyzed using volumetric adsorption isotherms. Based on the IUPAC isotherm classification (Fig. 2a–c and Supplementary Fig. 14a), the NMSCs exhibit a type-IV isotherm with the H3 hysteresis loops, indicating meso–macro porosity with wedge-shaped pores;[47] however, micropores cannot be detected by the t-plot method with a negative intercept (Supplementary Fig. 15).

The BET average surface areas of NMSC-1, NMSC-2, and NMSC-3 is 95, 124, and 226 $m^2 g^{-1}$ ($R^2$-0.999), respectively (Supplementary Table 2). By controlling $[Sn_2S_6]^{4-}$:$Mn^{2+}$ ratio, the porous structure formation pattern varies. We speculate that large surface area difference between NMSC-1 and NMSC-3 is related to the production of amorphous domains within the gel network where meso- and macropores are more likely forming via random aggregation. This is maybe the probable reason that NMSC-3, in which contains the most degree of amorphous domains, possess the largest BET surface area of all NMSCs. Large discrepancies within the surface area of each NMSCs explain the random aggregation nature sol–gel reaction. Pore-size distribution estimations using the Barrett–Joyner–Halenda (BJH) model support the specific porosity development indicated by the volumetric isotherm (Fig. 2b, d and Supplementary Fig. 14b). Both NMSC-1 and NMSC-2 exhibit well-developed meso-porosities, with interparticle pores characterized by pore diameters of 3.72–3.81 nm on the NMSC layer, whereas NMSC-3 exhibits no specific

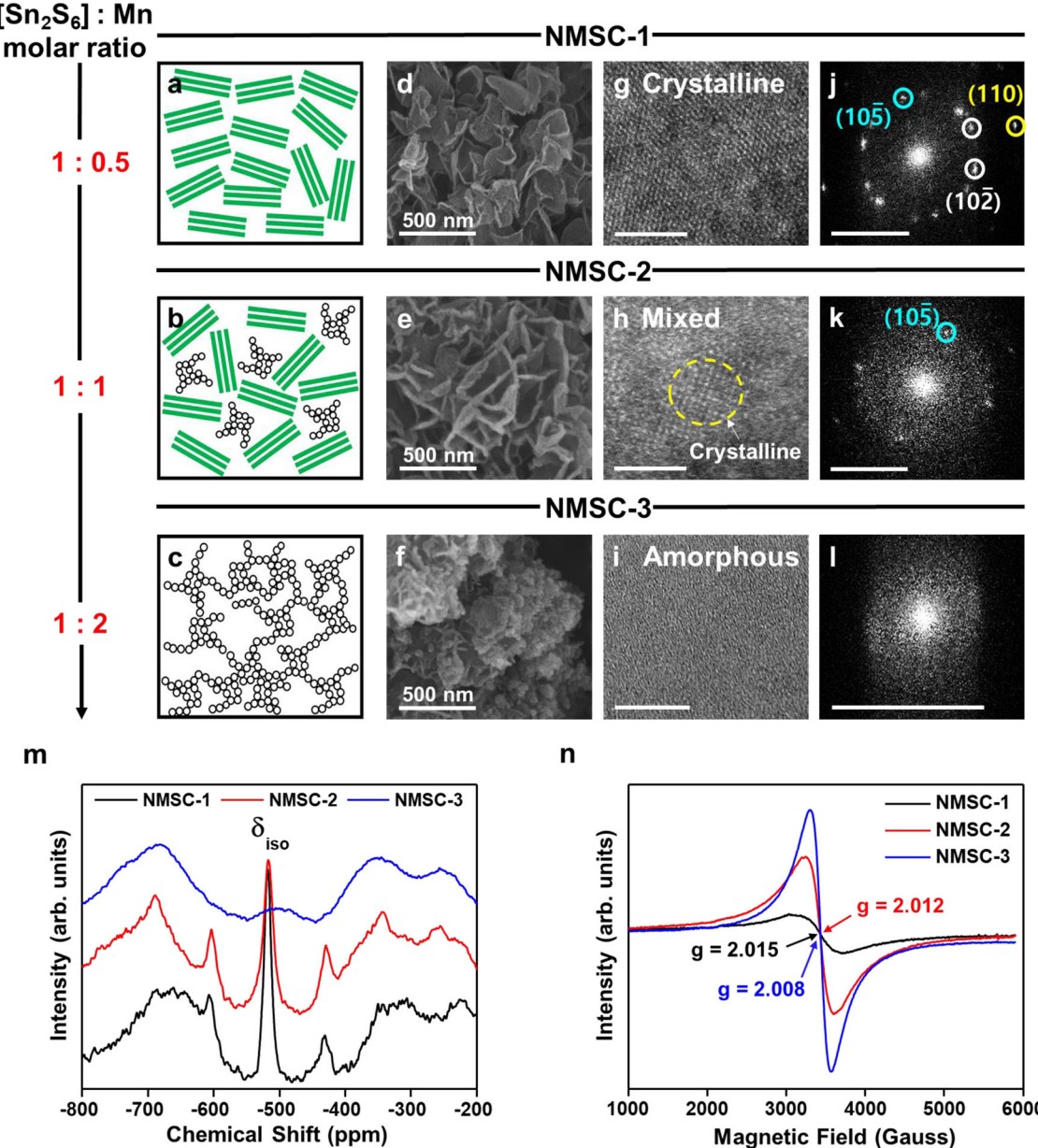

**Fig. 3 | Morphological and structural transformation control (layer–amorphous) for various stoichiometric NMSC ratios. a–c** Scheme and **d–f** field emission-scanning electron microscopy (FE-SEM), **g–i** scanning transmission electron microscopy (HR-STEM), and **j–l** selected area diffraction (SAED) images of NMSCs describing the structural transformation (layer–amorphous) of chalcogels at various [$Sn_2S_6$]$^{4-}$:$Mn^{2+}$ ratios. **a, d, g, j** NMSC-1 (1:0.5), **b, e, h, k** NMSC-2 (1:1), and **c, f, i, l** NMSC-3 (1:2) in 5 and 10 nm ranges. **m** Solid-state $^{119}$Sn-MAS NMR at 13 kHz. **n** Electro-paramagnetic resonance spectroscopy (EPR) confirming the octahedral coordination of the Sn center and unpaired electrons in the $Mn^{2+}$ and $Mn^{3+}$ centers with structural symmetry in the NMSCs. Source data are provided as a Source Data file.

mesopores[48]. The meso- and macropore development in NMSC-2 and NMSC-3 indicates that a rich manganese environment is more advantageous for promoting larger sized-pores than a manganese deficient environment. This matches with the aforementioned hypothesis that a low Mn content in [$Sn_2S_6$]:Mn results in a well-ordered layer-structure assembly in NMSC−1 with a highly ordered porosity, whereas a high Mn content in [$Sn_2S_6$]:Mn affords a more amorphous porous network with higher meso- and macro porosity in NMSC-3 development. Considering that the porosity in TACs-*n* with similar thiostannate-cluster constituents ([$Sn_2S_6$]$^{4-}$) can only be controlled by changing the ammonium counter-ligands[13], NMSCs have the unique feature of cluster-assembly-controlled porous structural variation with an apparent development of mesopores in the range 3.4–4 nm.

**Analysis of the morphological and structural changes in NMSCs**
To confirm the relation between the [$Sn_2S_6$]$^{4-}$: $Mn^{2+}$ stoichiometric ratio and the resulting macroscopic gel structures of all NMSCs (1, 2, and 3), the complimentary analyses of FE-SEM, HR-STEM, SAED were conducted. Depending on the various stoichiometric ratios, chalcogel is formed in crystalline (NMSC-1, Fig. 3a), mixed (NMSC-2, Fig. 3b), and amorphous (NMSC-3, Fig. 3c). The FE-SEM results of NMSCs (Fig. 3d−f) show a macroscopic gel-structure transformation with a stoichiometric increase of Mn in [$Sn_2S_6$]:Mn. Both NMSC-1 and NMSC-2 show a wrinkled-layer macrostructure, which indicates the aggregation of its two-dimensional constituents. We hypothesize that this wrinkle morphology is the result of o-MTC unit coagulation. At a [$Sn_2S_6$]:$^{4-}$Mn$^{2+}$ ratio of 1:0.5, the wrinkled-paper texture was more apparent, with the walls of the manganese-thiostannate layer structure sharing their basal

planes with the adjacent units, as observed in NMSC-1. Considering that graphene oxide-based porous structures often yield similar wrinkle morphologies, the macroscopic structure of NMSCs is expected to comprise two-dimensional layers[49]. As the Mn content in the $[Sn_2S_6]^{4-}$:$Mn^{2+}$ ratio increased, the layered morphology was replaced by an amorphous flake-shaped disordered morphology, as observed in NMSC-3. This suggests that the control of the $[Sn_2S_6]$:Mn stoichiometric ratio up to 1:1 maintains the degree of textural uniformity without sacrificing the layer-structure-based stacking nature of the NMSCs.

The nanostructure of the NMSCs can be observed via HR-STEM (Fig. 3g–i) and SAED (Fig. 3j–l). As expected, NMSC-1 with the lowest Mn content in $[Sn_2S_6]^{4-}$:$Mn^{2+}$ contained several domains of well-ordered lattices, with electron-diffraction spots indicating the specific planes of the $R\bar{3}m$ space group (Fig. 3j). Similar to PXRD analysis, the concentric circles of the NMSC-1 SAED pattern indicated lattice spacing at d = 0.28, 0.31, and 0.18 $nm^{-1}$ reflecting the (10$\bar{5}$), (10$\bar{2}$), and (110) planes, respectively. The reflections correspond to the Sn/Mn–S distance, where S is the projection of Sn between the (10$\bar{5}$) planes, the interlayer distance between the Sn or S atoms in the (10$\bar{2}$) orientation, and double-$d_{(110)}$ equal to the a-axis (Sn–Sn distance, 0.36 nm), respectively. In NMSC-2, these well-ordered domains of specific reflection appear less frequently. Although HR-STEM shows a sporadic emergence of ordered lattices, the SAED of NMSC-2 exhibits a partial development of specific reflection (e.g., (10$\bar{5}$), Fig. 3k). In contrast to NMSC-1 and NMSC-2, these specific lattice reflections disappear in NMSC-3. The HR-STEM image of NMSC-3 shows minimal signs of ordered domains; only an amorphous surface is revealed (Fig. 3i). SAED shows no specific diffraction pattern; conversely, a diffused ring is observed (Fig. 3l). A comparison of the HR-STEM and SAED patterns confirms that the variations in the $[Sn_2S_6]$:Mn stoichiometric ratio play a pivotal role in determining the micro− and macrostructure of NMSCs during gelation.

Furthermore, the HR-STEM images and SAED modes of the sample mixtures show a gradual phase development during gelation of NMSC-1 (Supplementary Fig. 16). At the thiostannate precursor ($Na_4Sn_2S_6$)-solution stage, three main diffractions correspond to the d-spacings of 0.18, 0.27, and 0.33 nm. The distance of 0.18 nm, maintained throughout the NMSC-1 gelation process, corresponds to $d_{(110)}$ in NMSC-1. This suggests that the intra-cluster distance of Sn−Sn is not affected significantly by the coordination transformations of the tin and manganese centers (Fig. 3j). A distance of 0.26−0.27 nm matches $d_{(10\bar{5})}$ in NMSC-1, reflecting the Sn/Mn−S distance (-0.257 nm of Sn−S bonding in KMS-1 crystallographic data)[29]. Considering the proposed coordination-transformation-induced gelation mechanism, the Sn/Mn−S distance is also maintained during the gelation process. In contrast, the diffraction peak of 0.33−0.34 nm, corresponding to the precursor solution, disappears over the gelation process in NMSC-1. This diffraction peak corresponds to an interlayer $d_{(10\bar{2})}$ orientation related to Na−S bonding, as interpreted from the reference peak of K − S bonding (-0.335 nm) from the original KMS-1 crystallographic data. When gelation is complete, the sodium atoms from the $Na_4Sn_2S_6$ cluster units are coordinated with the o-MTC units, which is possibly related to the development of the interlayer $d_{(10\bar{2})}$ orientation peak (d -0.31 nm). This suggests that the coordination transformation of MTC units occurs gradually without complete disintegration of the tetrahedral thiostannate cluster.

The $^{119}$Sn-MAS NMR spectrum of NMSCs provides further evidence for structural variation with the stoichiometric $[Sn_2S_6]$:Mn control (Fig. 3m). For NMSC-1, there is a single peak at $\delta_{iso}$ -520 ppm corresponding to the isotropic chemical shift of the octahedral $Sn^{4+}$ cluster (Supplementary Fig. 17)[50]. The absence of the 520 ppm peak from NMSC-3 suggests that a manganese-rich environment prevents the assembly of the tetrahedral MTC units into the ordered octahedral coordinated-layer structure of the o-MTC units. The other prominent

peaks between −200 and −700 ppm are the spinning sidebands of the chemical-shift anisotropic (CSA) contribution at 13 kHz spinning frequencies. The solid-state NMR signals of NMSC-3, unlike those of NMSC-1 and NMSC-2, suffered from a loss of sensitivity and resolution due to the hyperfine interaction[51]. This suggests a possible paramagnetic interaction arising from the metastable manganese center-induced disordered domains in the NMSC-3 structure.

The interaction of the unpaired electrons in $Mn^{2+}$ was analyzed by electron paramagnetic resonance (EPR) spectroscopy at room temperature (Fig. 3n), to gather a more localized information on the electron spin and surrounding geometry than NMR analysis. The solid-state EPR of NMSC shows an isotropic symmetry for low spin ($S=\frac{1}{2}$) signals separated by the two peaks. The powder line shape indicates that the magnetic moments in the NMSC samples are perpendicular to the direction of the external magnetic field. Consequently, the electron interactions in various d-orbitals are isotropic, indicating $g_x = g_y = g_z$. Thus, there is only one each of g-factor and external magnetic field where the resonance occurs. According to Supplementary Eqn. 1, the estimated g-factors are 2.015, 2.006, and 2.012 for NMSC-1, NMSC-2, and NMSC-3, respectively, which are similar to the g-factor of free electron ( $g_e = 2$–2.0023), indicating the manganese (II) $d^5$ complexes behavior of NMSCs[52]. The isotropic geometry of the NMSC EPR spectra exhibits typical octahedron or tetrahedron coordination complexes, with corresponding g-factors in a range 1.990–2.016[53]. The significant deviation of the EPR spectral signal intensities between the NMSCs suggests that a more ordered octahedral macroscopic structure (NMSC-1) yields a lesser degree of paramagnetic behavior than the mixed tetrahedral-octahedral coordination macrostructure (NMSC-3). This is consistent with the earlier hypothesis on the correlation between the central coordination mode and the resultant macrostructure ordering.

## Diversity of layered thiostannate chalcogels

Considering the diversity of stable-layered chalcogenides, a kinetic control between metathesis and coordination transformation can be generally implemented for the hierarchical aerogel synthesis with a layered chalcogenide structure. The synthesis of hierarchical aerogels in previous reports on $K_{2x}Mg_xSn_{3-x}S_6$ ($x = 0.5$–1, KMS-2) and $K_{2x}Sn_{4-x}S_{8-x}$ ($x = 0.65$–1, KTS-3) confirm the generality of the multiscale structure-control strategy for chalcogel synthesis[37,44]. As shown in Fig. 4a and d, PXRD confirms the structural similarity of Na-Mg-Sn-S ($Mg^{2+}$:$[Sn_2S_6]^{4-} = 0.5$:1, NMgSC-1) and Na-Sn(II)-Sn(IV)-S ($Sn^{2+}$:$[Sn_2S_6]^{4-} = 0.25$:1, NSn(II)SC-1) chalcogels, containing two-dimensional layered-structure motifs, to KMS-2 and KTS-3, respectively. To further understand the coordination transformation and structure switching of NMgSC and NSn(II)SC aerogels, complimentary structure analyses of these chalcogels are conducted through Raman spectroscopy and FE-SEM (Supplementary Figs. 18–19). Raman spectrum of NMgSC and NSn(II)SC samples exhibit three prominent bands at 257, 339, and 356 $cm^{-1}$. The $Sn_2S_2$ ring vibration leads to a peak at 257 $cm^{-1}$. The peaks at 339 and 356 $cm^{-1}$ are ascribed to Sn-S octahedral vibration in the tin cluster (Supplementary Fig. 18)[37]. The FE-SEM images reveal the morphological layer like flower flake for NMgSC-1 and NSn(II)SC-1 at low stoichiometric ratios. At high linker content, the layer morphology is damaged and switched to amorphous (NMgSC-2 and NSn(II)SC-2) (Supplementary Fig. 19). With kinetic control, Na-Mg-Sn-S and Na-Sn(II)-Sn(IV)-S show amorphous phase formations (Supplementary Fig. 20a, b) for fast gelation reactions at high $Mg^{2+}$ and $Sn^{2+}$ concentrations ($Mg^{2+}$:$[Sn_2S_6]^{4-} = 2$:1 for NMgSC-2 and $Sn^{2+}$:$[Sn_2S_6]^{4-} = 2$:1 for NSn(II)SC-2). Considering a similar mesopore formation (d ~4.0 nm for NMgSC-1 and 3.9 nm for NSn(II)SC-1) for the $Mg^{2+}$ and $Sn^{2+}$ counterparts, the development of small mesopores seems to be a general outcome of the stacking of the two-dimensional chalcogenide constituents (Fig. 4b, c, e, and f). As expected, NMgSC-2 and NSn(II)SC-

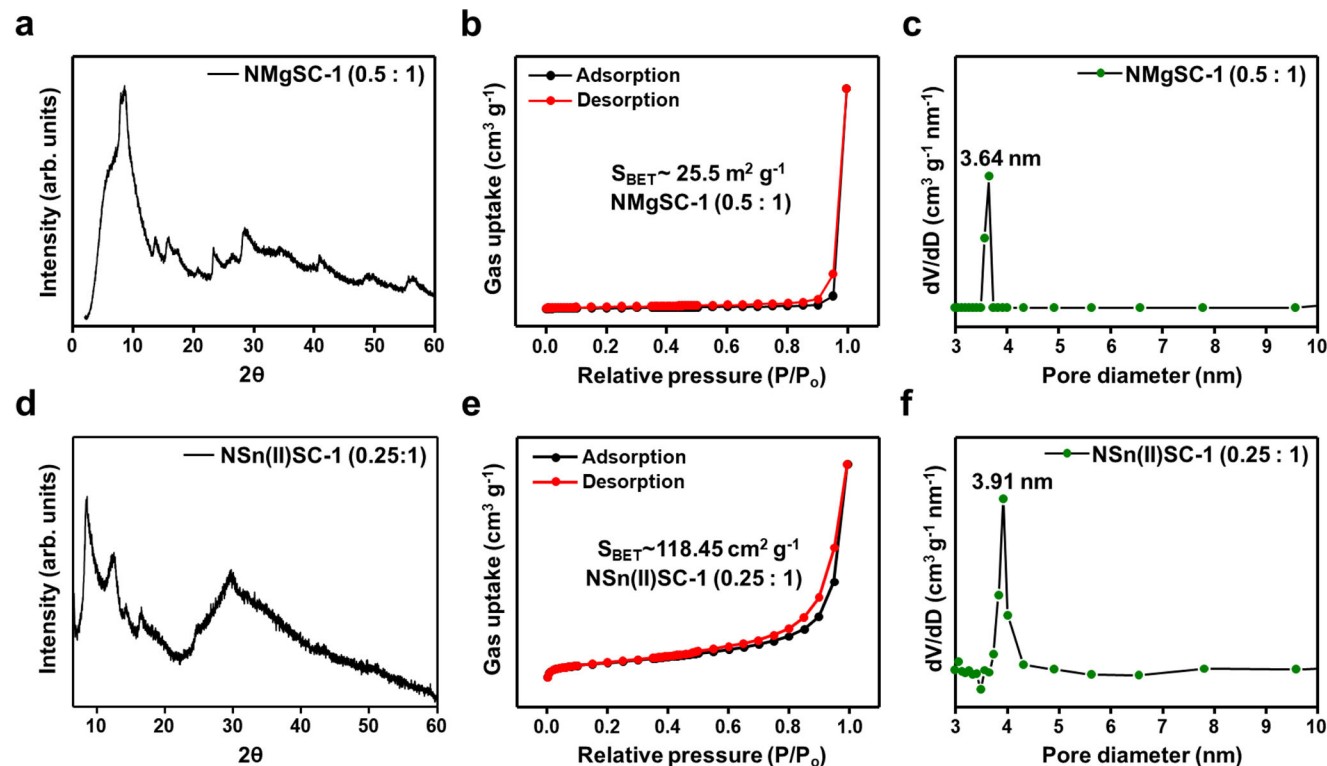

**Fig. 4 | PXRD and porosity analyses of layered thiostannate chalcogels with various linkers (Mg²⁺ and Sn²⁺). a** PXRD, **b** BET, and **c** BJH curves of Na-Mg-Sn-S ($Mg^{2+}:[Sn_2S_6]^{4-}$ = 0.5:1, NMgSC-1). **d** PXRD, **e** BET, and **f** BJH curves of Na-Sn(II)-Sn(IV)-S ($Sn^{2+}:[Sn_2S_6]^{4-}$ = 0.25:1, NSn(II)SC-1). Source data are provided as a Source Data file.

2 synthesized via fast metathesis reactions without coordination transformations exhibit a conventional porosity distribution (Supplementary Fig. 20c–f).

### Aqueous radionuclide-adsorption capabilities of NMSCs

To explore the potential functionality of NMSCs for effective adsorption, an aqueous $Cs^+$ remediation test was conducted. $Cs^+$ and $Sr^{2+}$ were selected as the targeting adsorbates due to their mid-range half-lives ($t_{1/2}$ = 30.2 and 28.8 years, respectively) and high fission yields (6.3 and 4.5%, respectively)[54]. Precedent examples of $Cs^+$ and $Sr^{2+}$ capture materials exist in the form of inorganic compound[29,55,56], carbon heterostructure[27,57,58], Prussian blue pigments[59–61], metal-organic-framework[62], and chalcogels[13], but most of them are only functional in complex heterostructure form or require multiple steps of synthetic protocols (Supplementary Table 3). In contrast, NMSCs are anticipated to be functional for $Cs^+$ and $Sr^{2+}$ capture through ion-exchange principle, as precedent layered chalcogenide compound (e.g., KMS-1) are known to attract them with high efficiency. Without further functionalizations, the soft basic surface and Na-ahchoring sites of NMSCs is anticipated to bound intermediate-to-soft cations of $Cs^+$ and $Sr^{2+}$, as other precedent chalcogenide aerogels have done successfully[13]. Figure 5a shows the $Cs^+$-removal capacity of the NMSCs. The adsorption result shows that the $Cs^+$-removal capacity varies among the NMSCs. NMSC-1 shows the highest average $Cs^+$-removal capacity (78 mg g⁻¹), whereas NMSC-2 and NMSC-3 show lesser $Cs^+$ uptakes (40 and 34 mg g⁻¹, respectively). This trend can be understood better if viewed from the relative sodium content of each NMSC; NMSC-3 having the highest manganese content and lowest sodium concentration ([Na]/([Sn]+[Mn]) = 0.06) exhibits the poorest $Cs^+$-removal capacity (34 mg g⁻¹) (Fig. 5b). The $Cs^+$ uptakes of NMSC-2 ([Na]/([Sn]+[Mn]) = 0.08) and NMSC-1 ([Na]/([Sn]+[Mn]) = 0.14) increase with sodium content to 40 and 78 mg g⁻¹, respectively. Therefore, we believe that the sodium in the aerogel structure plays a crucial role in the $Cs^+$ uptake via ion exchange through the hard–soft-

acid–base principle. The sulfide ion in NMSCs ($S^{2-}$), with a high electronegativity and polarizability, acted as a soft-base site, whereas the sodium ion ($Na^+$) with small ionic radius and low electronegativity acted as a hard acid. The seemingly unmatched sodium–sulfide pair, counterintuitively, exhibited a good ion-exchange functionality, rendering $Na^+$ a good leaving group upon dissolution in the aqueous medium. $Cs^+$, with a larger radius, lower positive charge, and soft-to-intermediate acidic tendency, could readily replace the sodium ion in an aqueous solution by forming stable interactions with $S^{2-}$. Furthermore, the layered structure, with several active sites at a low $Mn^{2+}$ content, contained the highly mobile sodium ions, which was advantageous for NMSC-1 to exhibit the dominant relative $Cs^+$-removal capacity. However, the divalent $Sr^{2+}$ cation was exchanged to a lesser degree than $Cs^+$ in all NMSC samples. The error bars in Fig. 5a represent the standard deviation of NMSC samples during $Cs^+$ and $Sr^{2+}$ ion exchange. The standard deviation of NMSC-1 for $Cs^+$ removal capacity fluctuates in a broader range (±26 mg g⁻¹) due to the amounts of $Na^+$ loss in each wet-gel synthesis. Other NMSCs illustrate a narrow standard deviation range (±7–11 mg g⁻¹), indicating the lesser ion-exchange activity. As NMSC-3 has the lowest $Cs^+$- and $Sr^{2+}$-removal capacity despite the highest surface area (226 m² g⁻¹), a well-defined pore structure within the layered structure and highly mobile sodium active sites of NMSC-1 and NMSC-2 have a greater influence on the resulting adsorption behavior than a mere enhancement of the surface area and pore volume and the presence of large (meso- and macro-) pores. To further understand the stability of chalcogels, the $Cs^+$ adsorption stability test is conducted. (Supplementary Fig. 21). The dried gel has a good wettability, so it is mechanically broken into smaller species in $Cs^+$ solution. During the equilibrium time (7 days), the physical shape of the gel is maintained. At the end of the stability test, chalcogel is converted to xerogel structure, leading to the pore collapse.

The adsorption behavior of NMSCs can be understood further through a kinetic analysis (Fig. 5c and Supplementary Fig. 22). As a

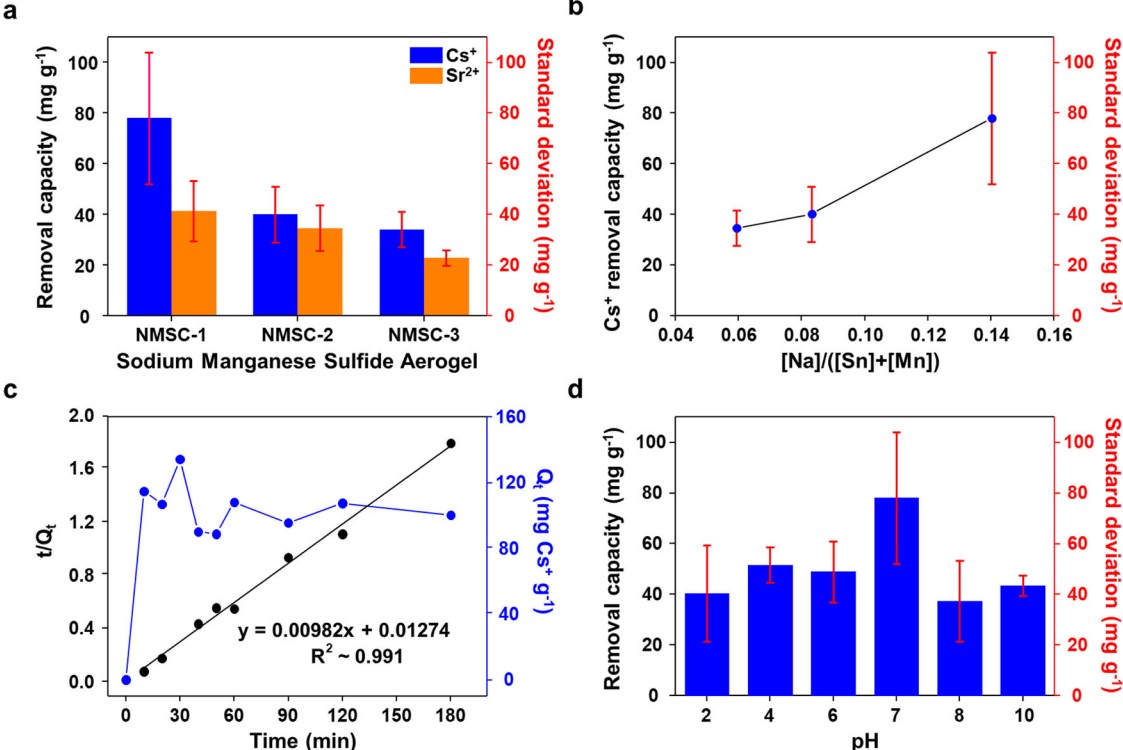

**Fig. 5 | Radionuclide remediation properties including error analysis of NMSCs.** **a** Radionuclide (Cs⁺, Sr²⁺)-removal capacities of NMSCs in an aqueous solution at pH = 7. **b** Cesium adsorption capacities of the NMSCs at different manganese ratios, revealing the sorption-behavior dependence on the Na content over the total metal content in the structure. **c** Raw adsorption kinetic (Q$t$ vs. time) of NMSC-1 for Cs⁺- exchanged (black line) and kinetic model of NMSC-1 described by pseudo-second-order fitting over the full equilibrium period (3 h) (blue line). **d** Cs⁺-adsorption capacity of NMSC-1 under various pH conditions. Error bars presents the standard deviation of the mean of three experiments. Source data are provided as a Source Data file.

representative, NMSC-1 shows rapid adsorption in the first 40 min period with a 100–140 mg g⁻¹ maximum adsorption capacity and saturation adsorption up to 400 min. As shown in the $t/Q_t$ vs. time plot, the pseudo-second order adsorption is dominant in the NMSC-1 kinetic model (Eq. 1)[63]. $Q_t$ and $Q_e$ are the Cs⁺-removal capacities (mg g⁻¹) of NMSC-1 at time $t$ and at equilibrium, respectively. $k$ (min⁻¹) is the rate constant of the pseudo-second-order model.

$$\frac{t}{Q_t} = \frac{1}{kQ_e^2} + \frac{t}{Q_e} \qquad (1)$$

Based on this kinetic model fitting of the sorption data over a 180 min period, NMSC-1 fits only the pseudo-second-order model ($R^2$ ~0.991, $Q_{e, cal}$ ~102, $k$ ~0.00757 (Cs⁺) and $R^2$ ~0.999, $Q_{e, cal}$ ~66.2, $k$ ~0.0059 (Sr²⁺)), indicating a chemisorption-oriented sorption behavior of the Na⁺–Cs⁺ and Na⁺-Sr²⁺ ion exchanges.

The competitive ion experiments were conducted to explore selective removal for Cs⁺ or Sr²⁺ on the marine-cation existing condition, representatively, Na⁺ (Supplementary Fig. 23 and Supplementary methods). The Cs⁺ removal capacity of NMSC-1 in the dual Na⁺–Cs⁺ solution decreased (from 4.7 to 11.3 mg g⁻¹) upon the Na⁺ concentration increase, while there is no significant difference in the Sr²⁺ removal capacity (14.1–15.1 mg g⁻¹) in the dual Na⁺ · Sr²⁺ solution. This trend demonstrated that divalent ion state (e.g., Sr²⁺) is less susceptible to the competitive ion effect than monovalent ion state (e.g., Cs⁺) in Na⁺-coexisting solution. This is another indirect evidence that NMSC control radionuclide through ion-exchange sorption.

Meanwhile, the unique morphology of NMSC-1 with layered structure can be examined using surface wettability measurement. The small water contact angle of NMSC-1 (θ ~ 11°) shows its hydrophilic surface with high surface energy. The high water wettability of NMSC-1

proposes the potential for candidate of Cs⁺ and Sr²⁺ adsorbent on the aqueous solution state (Supplementary Fig. 24 and Supplementary method).

The structural variation of the NMSC surface during the remediation process was tested using NMSC-1 as a representative. After the ion-exchange reaction with Cs⁺, NMSC-1 exhibited a significant change from layer to amorphous structure as indicated by PXRD and FE-SEM/EDS analyses (Supplementary Figs. 25–27). The XRD pattern of the ion-exchanged NMSC-1 shows broad peaks, with no peaks corresponding to the original NMSC-1 layer structure (Supplementary Fig. 25). Disappearance of signature peaks (5°, 9.8°, 19.8°, 28.7°, 33°, and 50.5°) from Cs⁺-exchanged and Sr²⁺-exchanged samples suggest that ion-exchange process transformed the original layer structure of NMSC-1 into amorphous structure. In addition, the FE-SEM analysis of ion-exchanged NMSC-1 show the transition from layer structure to amorphous state in which this change aligns well with the XRD result. This ion-exchange-induced amorphous transition demonstrate the flexible nature of NMSC structure upon the exposure to the external stimuli (Supplementary Fig. 26). This ion-exchange-induced phase control is not the focus of this study, but it would be an interesting theme of future study for potential connection to the field of reversible crystalline-to-amorphous phase transformation[64]. In addition, EDS mapping images and atomic % (Cs, Sr, Mn, Sn, and S) of NMSC-1 after Cs⁺ and Sr²⁺ adsorption exhibit the complete replacement of Na⁺ with Cs⁺ or Sr²⁺ after the ion exchange (Supplementary Fig. 27 and Supplementary Table 4), confirming the ion-exchange mechanism. Furthermore, the surface area of NMSC-1 after Cs⁺ ion exchange is measured by BET observing the changes on the surface area before and after ion exchange. The BET results show the collapsed surface area of NMSC-1 after Cs⁺ (2.3 m² g⁻¹) and Sr²⁺ (3.3 m² g⁻¹) ion exchanges (Supplementary Fig. 28). This demonstrates irreversible nature of ion-

exchange sorption of NMSCs. The chemisorption-induced ion-exchange behavior of NMSC is analyzed by XPS spectra and its atomic percentage. The results exhibit the content of Na$^+$ is fully replaced by Cs$^+$ (Supplementary Fig. 29 and Supplementary Table 5).

The influence of the charge on the NMSC surface during the adsorption is measured by the isoelectric point (point of zero charge, PZC). It is analyzed by the salt addition method (Supplementary methods)[65]. The $\Delta$pH was plotted against the initial pH values, and the isoelectric point is the pH at which $\Delta$pH is zero (Supplementary Fig. 30). In the pH 2-6 range, the $\Delta$pH values are positive with the maximum value at pH 3 (NMSC-1 and NMSC-2) and pH 4 (NMSC-3). The isoelectric points of NMSC are 7.13 (NMSC-1), 7.2 (NMSC-2), and 6.5 (NMSC-3) at 298 K, respectively. Based on the graph, the radionuclides adsorption (Cs$^+$ and Sr$^{2+}$) of NMSC should be better at the base solution by physical adsorption support of negative charge on the surface. However, the removal capacity of NMSC-1 showed its highest capacity obtained at a neutral charge surface. Therefore, we can conclude that chemisorption-induced ion exchange is the dominant radionuclide remediation principle. Adsorption stability at a wide range of pH is another unique feature of NMSC. At pH 2-10, NMSC-1 exhibited 37.4–56.1% ($\pm$4–26 mg g$^{-1}$ standard deviation) of the original Cs$^+$-removal capacity (Fig. 5d), indicating a potential ion-exchange competition with H$_3$O$^+$ under acidic conditions and hydroxyl group attacks on the sulfidic sites under basic conditions[66].

This study proposes unique coordination transformation and layer-stacking phenomena of thiostannate clusters: the underlying gelation principle of Mn-Sn-S bridge-based NMSC chalcogels. This new class of aerogels possesses the same microstructure as previously reported for layered metal sulfides (KMS-1), which are known for their extraordinary radionuclide (Cs$^+$)-remediation functionality via the hard–soft-acid–base (HSAB) ion-exchange principle. Sharing the common structural features of the anionic Mn-Sn-S bridge layer charge-balanced by alkali metal ions, NMSCs inherit its ion-exchange capabilities which can be used for an effective radionuclide control. In addition, the stoichiometric control of the Mn ion and thiostannate cluster allows structural transformation from the octahedral coordination-based uniform mesoporous channel to the tetrahedral coordination-based disordered random porous network. Furthermore, the successful synthesis of Na-Mg-Sn-S and Na-Sn(II)-Sn(IV)-S chalcogels, which share the structural features of NMSCs, demonstrates that the self-assembly of two-dimensional chalcogenide clusters can serve as an effective strategy for the design of exotic and uncommon hierarchical chalcogels.

# Methods
## Synthesis of Na$_4$Sn$_2$S$_6$·5H$_2$O
Na$_4$Sn$_2$S$_6$·5H$_2$O was synthesized from 14.410 g (60 mmol) of Na$_2$S·9H$_2$O (98%, JUNSEI) and 7.012 (20 mmol) of SnCl$_4$·5H$_2$O (98%, Sigma–Aldrich)[13]. Each salt was dissolved completely in 50 mL of deionized water, separately. Then, the SnCl$_4$·5H$_2$O solution was added dropwise to the Na$_2$S·9H$_2$O solution in an ice bath under vigorous stirring. The mixture was continuously stirred in the ice bath until the transparent solution obtained. After washing with 400 mL acetone, the white precipitate was collected by vacuum filter and dried under vacuum overnight.

## Synthesis of the NMSC chalcogels
The sodium manganese sulfide chalcogels were synthesized under an ambient atmosphere and denoted as NMSC-1, NMSC-2, and NMSC-3. The NMSCs were obtained via Mn$^{2+}$–Na$^+$ exchange in the presence of formamide at room temperature (25 °C). (CH$_3$COO)$_2$Mn·4H$_2$O (>98%, DEAJUNG) was dissolved in 4 mL of formamide, and 0.3 mmol of the corresponding tin sulfide precursor (Na$_4$Sn$_2$S$_6$·5H$_2$O ~232 mg) was added to 4 mL of formamide in a separate vial. The manganese-precursor solution was added slowly to the tin sulfide-precursor

solution with high-speed stirring and changed from a clear yellow solution to a yellow cloudy suspension depending on the Mn-precursor concentration. When the precursor solutions were completely mixed, a vigorous stirring (1 min) afforded a yellow cloudy suspension. The mixture samples were transferred and stored at room temperature for a week. To separate the remaining formamide, byproducts, and impurities, the solvent-exchange process was performed with a fresh supply of ethanol thrice a day for 2 weeks. The other gel systems such as Na-Mg-Sn-S (Mg$^{2+}$:[Sn$_2$S$_6$]$^{4-}$ = 0.5:1) and Na-Sn(II)-Sn(IV)-S (Sn$^{2+}$:[Sn$_2$S$_6$]$^{4-}$ = 0.25:1) were fabricated by adding magnesium (MgCl$_2$·6H$_2$O, ACS reagent, 99%, Sigma–Aldrich) or tin(II) additives (Sn(OAc)$_2$, Sigma–Aldrich) to the tin cluster via the stoichiometric ratio and synthesis procedure of the manganese gel.

## Synthesis of sodium manganese tin sulfide (NMS)
The polycrystalline NMS can be prepared with the optimized method of KMS-1 reported by Manos et. al. In typical synthesis, Sn (12 mmol), Mn (6 mmol), S (36 mmol), Na$_2$CO$_3$ (6 mmol), and H$_2$O (10 mL) were mixed in a 25 mL Teflon-lined autoclave at 250 °C for 4 days. The product was washed by distilled water (DI), carbon sulfide, and ethanol respectively. Finally, the obtained powder was dried under vacuum overnight.

## Supercritical-point drying of wet gels
The supercritical-point drying (CPD) method was used to replace the fluids in the wet gel with carbon dioxide at the supercritical point without structural destruction to maximize the specific surface area of the sample. A siphon-type liquid CO$_2$ tank was used for the CPD process. Initially, the wet gel was transferred from the vial and placed in a customized metal basket, which was then placed in the chamber of a custom-built dryer. Absolute ethanol was added until the sample was immersed. Then, liquid CO$_2$ was filled into the chamber at 4 °C for the exchange process. After 15 min of equilibrium, the mixture fluids were drained and replaced by fresh CO$_2$. The drain–refill process was repeated for 18 cycles during which the alcohol in the gel network was exchanged with liquid CO$_2$ step by step. In the final cycle, the temperature was raised to 40 °C and maintained for 10 min to ensure the transformation of liquid CO$_2$ to a supercritical fluid. The flow was then slowly reduced over 30 min until the pressure had dropped to 1 atm.

## Cs$^+$ and Sr$^{2+}$ removal test
For Cs$^+$ and Sr$^{2+}$ removal experiments, 100 ppm of 100 mL Cs$^+$ or Sr$^{2+}$ aqueous solutions were prepared using CsCl or SrCl$_2$, respectively. After stirring of 15 mg of NMSCs in 100 mL, 100 ppm CsCl or SrCl$_2$ aqueous solution for 3 h at 25 °C, all solutions were filtered for Cs$^+$ or Sr$^{2+}$ concentration detection.

## Cs$^+$ adsorption stability test
After placing dried NMSC samples in 2 mL Cs$^+$ solution (100 ppm), the samples are put in a stationary state without stirring during the consideration time (7 days).

## Solid-state $^{119}$Sn NMR analysis
Solid-state $^{119}$Sn magic angle spinning (MAS) NMR experiments were performed in the Bruker AVANCE II$^+$ 400 MHz NMR system (Bruker, USA) at the Seoul Western Center of the Korea Basic Science Institute (KBSI). To distinguish the real signal from the SSB (spinning side band), the MAS NMR experiments were conducted with spinning rates of 12 and 13 kHz. The analysis was performed with a 30 s delay time, one pulse sequence, p$_1$ = 2.1 (45°), and radio frequency = 149.1 MHz. For calibration, SnCl$_2$ (2.84 M) was dissolved in DMSO-d$_6$.

## Characterization
A volumetric isotherm analyzer (Autosorb-iQ-MP Quantachrome, USA) was used under degassing conditions for 90 min at 100 °C in a 9 mm

radius cell. The surface area was analyzed via $N_2$ adsorption. In addition, the BJH method was applied for analyzing the pore-size distribution. ICP-MS (iCAP RQ, Thermo Scientific, USA) and ICP-AES (OPTIMA 8300, PerkinElmer, USA) instruments were used to track the concentrations of $Cs^+$ and other metal ions in the NMSC-1 and NMSC-2-treated metal ion solutions. The samples were acidified using 0.015 mL of aq. $HNO_3$ and diluted 100 times. The acidification (0.02 mL $HNO_3$) and dilution (×100) were repeated to attain the required concentration for the ICP-MS and ICP-AES analysis. Raman spectra was obtained using LabRam Aramis Horiba with a special range of 100–3100 $cm^{-1}$ with 785 and 532 nm laser sources. Elemental analysis was conducted using FLASH 2000 (Thermo Scientific) and a thermal conductivity detector to obtain the weights of C, O, N, and S. Field emission-scanning electron microscopy (FE-SEM) was used to monitor the surface morphology of the chalcogels. The FE-SEM images of sodium manganese chalcogel were obtained using JEO USA JSM-6700F at the MEMS·Sensor Platform Center of Sungkyunkwan University. Transmission electron microscopy (TEM) was used to characterize the surface morphology of the chalcogels with an accelerating voltage of 200 kV. HR-STEM images were obtained using TITAN TM 80–300 (FEI, USA), and EDS/Mapping images using TALOS F200X (FEI, USA). Powder X-ray diffraction (PXRD) patterns of the chalcogel were obtained using D8-Advanced (Bruker-AXS, Germany) with a Cu K$\alpha$ radiation beam ($\lambda = 1.5406$ Å) in the 2θ range of 2–60° operating at 18 kW at room temperature. X-ray photoemission spectroscopy (XPS) data was obtained using ESCALAB 250Xi (Thermo-Scientific, UK) with monochromator Al K$\alpha$ (1486.6 eV). The SAXS patterns were measured using NANOPIX (Rigaku) with Cu K$\alpha$ radiation at the HANARO facility, KEARI, Republic of Korea. Silver behenate was used for data calibration. All the model fittings were carried out using the Irena/NIKA package[67].

## Data availability

The data that support the findings of this study can be found in the manuscript, Source Data file, and Supplementary Information or can be obtained from the corresponding authors upon request. Source data are provided with this paper.

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

## Acknowledgements

This study was supported by the National Research Foundation of Korea (NRF) and funded by the Ministry of Education (NRF-2020R1A2C4001617 to M.-G.K., NRF-2021M3H4A1A02055684 to M.-G.K.) and by the Technology Innovation Program (20004977) funded by the Ministry of Trade, Industry, & Energy (MOTIE, Korea). It was also funded by Air Environment

Complex Response programs (2E31663 to Y.O., 2E31711 to Y.O.) of the Korea Institute of Science and Technology (KIST).

## Author contributions

T.D.C.H. and H.L. conceived the idea under the supervision of M.-G.K. and Y.O. T.D.C.H. performed material synthesis, gelation mechanism analysis, and ion exchange experiements. H.L. performed porosity characterization, gelation mechanism analysis, and ion exchange experiements. T.D.C.H., H.L., Y.O., and, M.-G.K. performed data analysis. H.L. and Y.O. performed mechanistic TEM and data analysis. H.L., Y.O., B.K., and I.C. assisted BET, TEM, and ICP analyses. Y.K.K. and K.A. assisted solid-state 119Sn NMR and PXRD measurement. H.M.J. performed SAXS and data analysis. T.D.C.H., H.L., M.-G.K., and Y.O. wrote the manuscript. M.-G.K. and Y.O. supervised the project. All authors contributed to the final version of the manuscript.

## Competing interests

The authors declare no competing interests.
