## [Peer Review File · Nature Communications]

Multiscale structural control of thioannate chalcogels with two-dimensional crystalline constituentsREVIEWER COMMENTS

Reviewer #1 (Remarks to the Author):

Recommendation: Publish in Nature Communications after major revisions

This article reports a self-assembly strategy to synthesize precise multiscale structure of chalcogels. The aerogels exhibited excellent phase crystallinity and hierarchical porous structure, which shown selective radionuclide (Cs^+ , Sr^{2+})-control functionality. This strategy is interesting, the data is substantial and the synthesis principle is also clearly elaborated. Thus, this manuscript can be published in Nature Communications after addressing the following points:

1. In the process of synthesizing NMSC aerogels, the raw material used is $(\text{CH}_3\text{COO})_2\text{Mn}\cdot 4\text{H}_2\text{O}$, where the valence of Mn is +2. While Mn in NMSC is a mixed-valence state of Mn^{2+} and Mn^{3+} , what is the reason for this? Will a mixed-valence state of Mn^{2+} and Mn^{3+} have some effect on the formation of crystalline NMSC?
2. As mentioned in the manuscript, there is a crystal structure of NMSC (Supplementary Figure 3), and how this structure is obtained? Since I do not see the presence of Mn in the crystal structure, where is Mn located in the crystal? If the structure is not the crystal structure of an NMSC, even if the structure is similar, it cannot be presented as NMSC data.
3. In the manuscript, only the morphologic structure of the synthesized NMSC is shown. To my way of thinking, the morphologic structure of the initial sample (KMS-1) should be shown to compare the morphologic changes before and after self-assembly strategy, to further make sure the successful synthesis of NMSC.
4. As mentioned in line270, "NMSCs exhibit a type-II isotherm have the H3 hysteresis loops, indicating meso-and macro porosity with wedge-shaped pores", This is not accurate, which Type-II isotherm does not generally form hysteresis loops. Please carefully check and compare the literature to redetermine the adsorption type.
5. It is mentioned that the removal capacity of Cs^+ is related to the content of Na^+ in NMSC, which the chemisorption-oriented sorption behavior is by the Na^+ - Cs^+ ion exchange. So, the content of Na^+ in the samples before and after removing Cs^+ should be compared?
6. After removing Cs^+ , the crystallinity and morphological characteristics of the samples are significantly changed, and what is the reason for this phenomenon? After Na^+ - Cs^+ ion exchange, 400-4000 nm holes are formed. What is the cause of the formation? Is it the collapse of the internal structure caused by ion exchange or this ion exchange can form large pores in a controllable manner? It would be better to illustrate this by an adsorption/desorption experiment.
7. Some published papers closely related to the present work are suggested to be cited and commented such as Chem. Mater. 2013, 25, 2116–2127; Chem. Sci., 2016, 7, 1121–1132; J. Mater. Chem. A, 2016, 4, 16597–16605; Materials Lab, 2022 1, 220046.

Reviewer #2 (Remarks to the Author):

The manuscript by Kim and co-authors reported a multiscale structural control method through the thioostannate motif-transformation-induced self-assembly process. The chalcogels fabricated in this work with a layer-by-layer stacking structure showed the adsorption performance of cesium and strontium ions. It is an interesting work, and the reported method may contribute to the design of unique chalcogels with different structural hierarchies. However, many important issues need to be addressed before the manuscript can be further considered for publication.

1. The formation mechanism of the layered and amorphous structures is important for this work, and the illustration shown in Figures 2a-b could not be seen clearly. For readers' better understanding, I would recommend a reorganization of Figure 1 and Figure 2. Figures 2a-b are recommended to be enriched and placed in Figure 1.
2. Figure 5d showed the trend of the Cs⁺-adsorption capacity of NMSC-1 under various pH conditions. It seems that the best data might be in the range of pH = 4 ~ 6. However, the data at pH = 7 have the best performance of adsorption. Thus, the additional adsorption capacity data between pH = 6 and 8 could be good support.
3. What is the coordination structure of various linkers like? Do the chalcogels have a layered structure? The corresponding characterization should be supplemented.
4. How is the stability of chalcogels during the adsorption process? Will the chalcogels swell? The long-time running and stability test for adsorption should be explored.
5. The authors should comment on why chose cesium and strontium for exploring the application potential of chalcogels? What are the advantages of chalcogels in this work compared to other cesium and strontium adsorption materials?
6. The adsorption of cesium and strontium in the current work cannot fully reflect the structural advantages of chalcogels. Can the chalcogels in this work capture cesium and strontium selectively? Do chalcogels have superior performance for the separation and adsorption of other ions, like uranium?
7. In the current manuscript, only the adsorption kinetic of chalcogels for cesium was discussed, the adsorption kinetic of chalcogels for strontium should also be added.
8. To help readers understand this field, the recent progress in the selective separation and adsorption of strontium and cesium ions from aqueous solutions should be mentioned in the introduction.
9. The characterization (e.g., surface charge) for NMSC should be added and further discussed. Optical images of the aerogel and its wet-gel state should be modified because they cannot reflect the similarity of state and volume mentioned in the article.
10. The snapshots for water contact angle on the surface of chalcogels should be added. In this work, the authors proposed that chalcogels can be used for aqueous radionuclide-adsorption, and the surface wettability is important for the adsorption ability.
11. In general, the error analysis of the chalcogels should be added, especially in the experiments on the adsorption capacity of radial ions. Also, the necessary labels of the y-axis should be complemented.

Reviewer #3 (Remarks to the Author):

The manuscript titled “Multiscale structural control of thiostannate chalcogels with two-dimensional crystalline constituents” presents the synthesis, structural characterization, and ion exchange properties of a new family of Mn based chalcogenide aerogels exhibiting KMS-1 like layered elements. The authors claim that the porosity, macrostructure, crystallinity, and kinetics are sensitive to the Sn:Mn ratio in the precursor solutions. The authors go on to support these claims by preparing three distinct materials utilizing this method of control; they are then extensively studied via an array of spectroscopic and scattering techniques. Their findings are important and of broad interest to the materials chemistry, material science, and ion exchange communities. A major finding of this work is the control of structure which may enable a hierarchical structural design of future chalcogel materials that has been limited by existing synthetic techniques. While the ion exchange characterization of these materials is enough to imply their ability to ion exchange, more studies are needed to understand their ion exchange performance fully. Without both a detailed study on the ion exchange capabilities of these materials and evidence that the synthetic control they have developed can be used to enhance this capability, it is difficult to justify the acceptance of this manuscript to this journal. Overall, this manuscript presents an important and potentially impactful contribution to the existing literature on chalcogenide aerogels and is recommended for publication in the Journal **Nature Communications** following the major revisions detailed below.

Prior to publication, the authors should address the following major issues.

- 1) Figure 2a-b is somewhat misleading to the reader due to the depicted coordination environment of Mn (square planer to my eye). The authors do not comment on the coordination of Mn in the text beyond correctly stating that it prefers octahedral environments. This figure thus implies a mechanistic understanding of the assembly which is not supported by the data presented. It would be more appropriate to depict the interaction of Mn with the $[\text{Sn}_2\text{S}_4]^{-4}$ cluster using exclusively arrows to communicate only chelation and not coordination. In addition, because there is no real mechanistic data both the t-IMTC unit and the image depicting the transformation should be removed/altered. Such a reaction scheme should be constructed using pictures, not bonds, to avoid implying a local understanding of this process. The image below shows the interactions I’m referring to circled in red along with an example of how the chelation interaction could be drawn.

- 2) On line 272 it states “*The BET surface areas of NMSC1, NMSC-2, and NMSC-3 increase with the Mn content in [Sn₂S₆]:Mn to 115, 115, and 250 m² 273 g⁻¹ (R² 274 ~0.999), respectively.*” Why do NMSC-1 and NMSC-2 have the same surface areas, given their differences in structure? Shouldn't the surface area of NMSC-2 > NMSC-1?
- 3) The BET surface areas should be measured both before and after ion exchange to observe changes in the surface area that come from this process.
- 4) EDS-SEM analysis should also be conducted before and after the exchange to confirm loading of the Cs/Sr in the materials and the absence/reduction of Na in the compounds. Figure S17 depicts the SEM micrographs of the materials post ion exchange but does not show the EDS of the materials as well.
- 5) The authors should include a comparison of the performance of their materials with other Cs sorbent materials present in the literature to better contextualize their results.
- 6) Competitive ion exchange studies should be conducted with the materials to understand their selectivity and enable comparison with existing materials.

In addition to the above Major revisions, the authors should also address the following minor revisions:

- 1) The title should be changed to communicate the ion exchange experiments done with these materials.
- 2) Line 55 “adamantine clusters ([M₄S₁₀]⁴⁻, M = Sn, Ge/Q = S, Se)” I think it is Q instead of S.
- 3) On Lines 32, 105, 407, 479 . The word “selective” should be removed. It is not appropriate to describe these materials as selective as competitive experiments (i.e. Cs vs Sr in the same solution) have not been conducted.
- 4) Beginning on line 421 the authors state “*The seemingly unmatched sodium–sulfide pair, counterintuitively, exhibited a good ion-exchange functionality, rendering Na⁺ a good leaving group upon dissolution in the aqueous medium.*” in reference to the performance of these materials in exchanging Na⁺ ions for Cs⁺ ions. Describing this observation as counterintuitive is confusing as the “*unmatched sodium-sulfide pair*” is the reason why this material is able to exchange in the first place, which would make the performance *intuitive*. I ask the authors to clarify the wording of this sentence.
- 5) Beginning on line 465, the authors state “*At pH 2–10, NMSC-1 exhibited 37.4–56.1% of the original Cs⁺-removal capacity (Fig. 5d), indicating a potential ion-exchange competition with H₃O⁺ under acidic conditions and hydroxyl group attacks on the sulfidic sites under basic conditions. This is a **unique** feature of NMSCs, which was not observed in the original layered metal sulfide (KMS-1) structural motif.*” This statement is untrue as in the original report of KMS-1 (ref 27 in the manuscript), the variable pH Cs ion exchange study conducted demonstrated removal percentages of ~42-76% of the theoretical capacity over the pH range of 1-12. Furthermore, it is imprecise to describe these results by saying “*At pH 2–10, NMSC-1 exhibited 37.4–56.1% of the original Cs⁺-removal capacity*” as this experiment does not determine the capacity of the material at different pH's, these are more precisely removal percentages. To make this statement

one would need to conduct isotherms at each pH of interest to comment on the change in capacity over pH. I suggest the authors rephrase their statement to something like “...exhibited removal percentages of 37.4–56.1% in reference to the capacity determined at pH=7”.

We thank the reviewers for their helpful advices, which strengthen the scientific importance of our work. In the following, we addressed point by point all the comments made by the reviewers to our manuscript (NCOMMS-22-28388). All revisions to the manuscripts addressing the comments of the Reviewers are highlighted in the blue text.

Response to the Reviewer 1's comment in the first review

[Comment]

Recommendation: Publish in Nature Communications after major revisions

This article reports a self-assembly strategy to synthesize precise multiscale structure of chalcogels. The aerogels exhibited excellent phase crystallinity and hierarchical porous structure, which shown selective radionuclide (Cs^+ , Sr^{2+})-control functionality. This strategy is interesting, the data is substantial and the synthesis principle is also clearly elaborated. Thus, this manuscript can be published in Nature Communications after addressing the following points:

[R1, Comments #1]

In the process of synthesizing NMSC aerogels, the raw material used is $(\text{CH}_3\text{COO})_2\text{Mn}\cdot 4\text{H}_2\text{O}$, where the valence of Mn is +2. While Mn in NMSC is a mixed-valence state of Mn^{2+} and Mn^{3+} , what is the reason for this? Will a mixed-valence state of Mn^{2+} and Mn^{3+} have some effect on the formation of crystalline NMSC?

[R1, Response #1]

Thank you for your kind comment. As pointed out, we observed the partial oxidation of Mn^{2+} into mixed valance state of $\text{Mn}^{2+}/\text{Mn}^{3+}$. As reported, the ion exchange of KMS-1 with larger size cations, such as Rb^+ and Cs^+ , resulted layer distance increase (from 8.47 Å to 8.65 Å and 8.96 Å, respectively) and mixed valance state of $\text{Mn}^{2+}/\text{Mn}^{3+}$, regardless the existence of ambient O_2 ²⁹. Although the ambient exposed Cs^+ exchanged KMS-1 shows larger portion of Mn^{3+} , the ion exchange of Cs^+ under N_2 still result the significant oxidation of Mn^{2+} into Mn^{3+} ($\text{Mn}^{2+}:\text{Mn}^{3+} = 1:1.1$). Overall the mixed valance state is more thermodynamically stable state

for KMS-1 structure with larger layer distance after ion exchange. Considering the larger layer distance of NMSC-1 (9.02 Å) and the solution phase synthesis condition under ambient air, which is similar to ion exchange reaction of KMS-1 under ambient condition, the mixed valance of Mn²⁺/Mn³⁺ with partial oxidation of Mn²⁺ might be more thermodynamically stable structure. Changes are listed as following:

✓ **Mixed valence state of Mn²⁺ and Mn³⁺ (L230-234)**

“Moreover, considering that the typical splitting-energy separations (ΔE) of Mn 3s are 5.8–6.2 eV for Mn(II) and 5.2–5.5 eV for Mn(III), NMSC-1 exhibits a mixed-oxidation state of Mn²⁺ and Mn³⁺, with a ΔE of 5.6–5.7 eV for Mn 3s (Supplementary Fig. 10)^{42, 43}. Based on the study of ion-exchange KMS-1, the mixed-valence state of Mn²⁺ and Mn³⁺ existed stably in crystalline structure with a larger interlayer distance investigated in ambient air or N₂ gas²⁹. Therefore, the solution phase synthesis of NMSC with similar layer distance could result in the mixed valance state of Mn²⁺ and Mn³⁺ as a thermodynamically stable phase for NMSC-1.”

[R1, Comment #2]

As mentioned in the manuscript, there is a crystal structure of NMSC (Supplementary Figure 3), and how this structure is obtained? Since I do not see the presence of Mn in the crystal structure, where is Mn located in the crystal? If the structure is not the crystal structure of an NMSC, even if the structure is similar, it cannot be presented as NMSC data.

[R1, Response #2]

Thank you for your kind comment. First, we apologize for our mistake in the original Supplementary Figure 3, which should show Sn/Mn mixed site instead of Sn site. We fixed the figure accordingly. Moreover, according to your suggestions, we measured and compared the PXRD of NMS powder to KMS-1 material. In addition, we revised the Sn/Mn position in crystal structure. All the results are added in the main manuscript and supplementary information (SI). To confirm the crystal structure of NMSC, we synthesized the sodium manganese tin sulfide (NMS) bulk material comparing to KMS-1 material via X-ray diffraction. As a result, the powder X-ray diffraction (PXRD) peak of NMS shows almost similar peak positions to KMS-1 material. Therefore, we conclude that the crystal structure of NMSC is

totally matched with the KMS-1 material. In addition, we explained the crystal structure formation and revised the Sn/Mn position in the scheme which shows the representative peak positions of NMSC-1 as shown below.

✓ SI Figure 5-6 updates

Supplementary Figure 5. X-ray diffraction of Sodium manganese tin sulfide (NMS) powder material.

Supplementary Figure 6. $R\bar{3}m$ crystal structure. The specific representative planes of NMSCs according to the peak positions in X-ray diffraction (XRD) of sodium manganese tin sulfide powder (NMS) and KMS-1 materials.

✓ NMS powder material (L151-155 and L158-161)

“The main peaks of NMSC-1 resemble the Bragg diffraction peaks (10.1°, 20.3°, 28.7°, 33.0°, and 50.5°) of $K_{2x}Mn_xSn_{3-x}S_6$ (KMS-1), a well-known layered metal sulfide (Supplementary Fig. 4)²⁹. Furthermore, the crystal structure pattern of NMSC-1 is well confirmed by the PXRD patterns of NMS and KMS-1 bulk materials (Supplementary Fig. 5).”

“The prominent peaks of NMSC-1 at 9.8° and 19.8° 2 θ are the representative (003) and (006) reflections of c-axis lengths in the $R\bar{3}m$ space group (Supplementary Fig. 6)³⁰, corresponding to an interlayer distance of 9.02 Å and a c-axis length of 27.06 Å (by multiplying 3d₍₀₀₃₎ in NMSC-1.”

✓ KMS-n structure description (L246-253)

“To confirm the two-dimensional local structural motif of NMSC, KMS-1, the mother structure of NMSCs, is synthesized and characterized through SEM, and EDS analysis. (Fig. 1d). The typical metal layer sulfides ($KMSn_2S_6$) families) in the space group of $R\bar{3}m$ includes KMS-1 (M = Mn²⁺)²⁹, KMS-2 (M = Mg²⁺)⁴⁴, and KMS-5 (M = In³⁺)⁴⁵. The layer is constructed by (M/Sn)S₆ octahedral coordination with M and Sn atoms occupying the same position. These structures are generally obtained by high temperature and pressure hydrothermal synthesis. We anticipate that NMSCs develop the structural motif of KMS-1 with the Na⁺ replacement at the K⁺ position during the spontaneous gelation.”

[R1, Comments #3]

In the manuscript, only the morphologic structure of the synthesized NMSC is shown. To my way of thinking, the morphologic structure of the initial sample (KMS-1) should be shown to compare the morphologic changes before and after self-assembly strategy, to further make sure the successful synthesis of NMSC.

[R1, Response #3]

We agree with the reviewer's comment. According to your suggestions, we redrew and added the morphologic structure and EDS mapping, including K, Mn, Sn, and S elements of the initial reference sample (KMS-1) in Figure 1d, as shown below. Now, Fig. 1 contain gelation process, EDS mapping, XRD, Raman, XPS, KMS-1 structure and SEM/mapping, and Mn:[Sn₂S₆]⁴⁻

ratio-induced gelation pattern distinction, providing an exclusive overview of the NMSC synthesis result. In Figure 1d, more detailed structural description of KMS-1 is added to confirm the structural similarity with NMSCs. Figure 1e presents Mn:[Sn₂S₆]⁴⁺ ratio-induced gelation pattern distinction of NMSC-1 (low Mn ratio) and NMSC-3 (high Mn ratio) which are representative example of slow and ordered or fast and disordered gelation, respectively. Also, gelation progress details are added for all NMSCs in SI Fig. 1 to show distinctive gelation patterns based on stoichiometric Mn:[Sn₂S₆]⁴⁺. Changes are listed as following:

- ✓ Stoichiometric variation-induced structure formation (L256-271)

“All three NMSCs present a common metathesis reaction with a donor-acceptor bond between Mn²⁺ and tetrahedral thioannate cluster (t-[Sn₂S₆]⁴⁺). If slow gelation allows thermodynamically stable phase formation, as in the case of NMSC-1 formation, the manganese center favors a densely packed octahedral coordination and causes a shift in the core coordination of the Sn⁴⁺ center from tetrahedral to octahedral. Resulting octahedral coordinated manganese thioannate cluster triggers layer-by-layer stacking aggregation to form the layered NMSC-1 structure, as shown in Fig. 1e (1). However, increasing the molar ratio of [Sn₂S₆]:Mn to 1:1 (NMSC-2) or 1:2 (NMSC-3) stimulates the gelation process, leading to a mixed tetrahedral-octahedral coordination formation of the Sn⁴⁺ center. The abundant stoichiometric metal linking centers (Mn²⁺) facilitated a rapid assembly of mixed Sn⁴⁺ coordination. As a result, the mixed coordination of Sn⁴⁺ center creates a three-dimensional amorphous macrostructure with short-range ordering and random orientation. This random self-cross-linking behavior becomes more dominant as the stoichiometric Mn²⁺ ratio increase, as in the case of NMSC-3 (Fig. 1e (2)). Thus, we speculate that the stoichiometric-Mn²⁺-variations play a crucial role in determining the coordination mode of NMSC and its corresponding porous structure.”

✓ Figure 1 reformation (L136-144)

Fig. 1 Synthesis, characterization, and hypothesized mechanism of coordination (Tetrahedral-Octahedral)-structure (Layer-Amorphous) switching of sodium manganese sulfide chalcogel (NMSC) with a layered structure similar to that of metal sulfide (KMS-1) materials. **a** NMSC synthesis. **b** EDS mapping of layered NMSC-1, including Na, Mn, Sn, and S. **c** Structural analyses of NMSC-1 using PXR, FT-IR, and XPS analyses of NMSC-1 for Sn 3d and Mn 2p respectively. **d** Scheme and EDS mapping of layered KMS-1 structure,

including K, Mn, Sn, and S. e Hypothesis mechanism of coordination (Tetrahedral-Octahedral) and structure (Layer-Amorphous) transformation in NMSC.

✓ Gelation of NMSCs (L120-125)

“As shown in Fig. 1a, a transparent yellow precursor solution containing Na^+ , Mn^{2+} , and $[\text{Sn}_2\text{S}_6]^{4-}$ turned to a light-amber turbid solution over 5h, and subsequently to a darker wet gel whose viscosity increased until a mechanically integrated monolithic gel was formed in 7 d. With subsequent solvent-exchange and critical point drying (CPD) processes, a monolithic and homogeneous light-colored aerogel was produced, retaining the volume of its wet-gel state (Fig. 1a and Supplementary Figs. 1-2).”

Supplementary Figure 1. Gelation process of NMSC samples. (a) As prepared, (b) 15 min, (c) 12 h, and (d) 7 days.

Supplementary Figure 2. Optical images of (a, e) NMSC-1, (b, f) NMSC-2, and (c, g) NMSC-3 before and after critical point drying (CPD).

In the optical images of NMSC samples (Supplementary Fig. 2), the rigidity of wet gel was directly proportional to the added amount of manganese. Overall, the physical shape of wet gel was maintained after critical point drying process.

✓ KMS structure discussion (L246-253)

“To confirm the two-dimensional local structural motif of NMSC, KMS-1, the mother structure of NMSCs, is synthesized and characterized through SEM, and EDS analysis. (Fig. 1d). The typical metal layer sulfides (KMSn_2S_6) families) in the space group of $R\bar{3}m$ includes KMS-1 ($M = \text{Mn}^{2+}$)²⁹, KMS-2 ($M = \text{Mg}^{2+}$)⁴⁴, and KMS-5 ($M = \text{In}^{3+}$)⁴⁵. The layer is constructed by $(M/\text{Sn})\text{S}_6$ octahedral coordination with M and Sn atoms occupying the same position. These structures are generally obtained by high temperature and pressure hydrothermal synthesis. We anticipate that NMSCs develop the structural motif of KMS-1 with the Na^+ replacement at the K^+ position during the spontaneous gelation.”

[R1, Comments #4]

As mentioned in line270, "NMSCs exhibit a type-II isotherm have the H3 hysteresis loops, indicating meso-and macro porosity with wedge-shaped pores", This is not accurate, which Type-II isotherm does not generally form hysteresis loops. Please carefully check and compare the literature to redetermine the adsorption type.

[R1, Response #4]

Thank you for the comment. We have revised the corresponding area with correct designation of isotherm type as following:

- ✓ Isotherms type correction (L289)

“... the NMSCs exhibit a type-IV isotherm with the H3 hysteresis loops...”

[R1, Comments #5]

It is mentioned that the removal capacity of Cs^+ is related to the content of Na^+ in NMSC, which the chemisorption-oriented sorption behavior is by the Na^+ - Cs^+ ion exchange. So, the content of Na^+ in the samples before and after removing Cs^+ should be compared?

[R1, Response #5]

It is reasonable point out the necessity of tracking down the content of Na^+ to confirm our speculation that Cs^+ is removed from the water source by the ion-exchange principle. As suggested, we have conducted ICP and XPS analysis of pristine and ion-exchanged NMSC gels. In addition, we added the XPS spectra of NMSC samples before and after Cs^+ ion exchange which exhibits the content of Na^+ is fully replaced by Cs^+ . We also described the Na^+ - Cs^+ ion exchange of NMSC samples in the main manuscript and added the analysis results in supplementary information (SI).

- ✓ XPS result of ion-exchanged NMSCs (L545-547)

“The chemisorption-induced ion-exchange behavior of NMSC is analyzed by XPS spectra and its atomic percentage. The results exhibit the content of Na^+ is fully replaced by Cs^+ (Supplementary Fig 29 and Supplementary Table 5).”

Supplementary Figure 29. XPS spectra of NMSC samples before and after Cs⁺ exchange. (a) NMSC-1, (b) NMSC-2, and (c) NMSC-3.

Supplementary Table 5. Atomic % by XPS analysis of NMSC samples before and after Cs⁺ ion exchange.

Sample	Name & Atomic %						
	Na1s	Cs3d	Mn2p	O1s	Sn3d	C1s	S2p
NMSC-1	3.07	-	2.39	20.67	11.85	46.61	15.42
Cs ⁺ exchanged	-	3.02	2.48	21.65	13.85	34.63	24.36
NMSC-2	3.09	-	3.65	23.42	13.72	34.24	19.60
Cs ⁺ exchanged	0.52	3.19	2.47	17.96	14.05	44.54	17.28
NMSC-3	1.58	-	5.22	29.00	11.96	38.15	14.09
Cs ⁺ exchanged	0.1	1.23	3.58	27.28	18.02	32.23	17.56

[R1, Comments #6]

After removing Cs⁺, the crystallinity and morphological characteristics of the samples are significantly changed, and what is the reason for this phenomenon? After Na⁺-Cs⁺ ion exchange, 400-4000 nm holes are formed. What is the cause of the formation? Is it the collapse of the internal structure caused by ion-exchange or this ion exchange can form large pores in a controllable manner? It would be better to illustrate this by an adsorption/desorption experiment.

[R1, Response #6]

Thank you for your kind comment. According to your suggestions, we revealed the change of crystallinity and morphological characteristics of NMSC samples in the main manuscript. In the same context as comment #5, it is reasonable to monitor the changes of NMSCs after the ion-exchange process to understand the basic nature of this materials. The transformation of NMSC phase is due to the surface tension arise from the diffusion and contact of water on the surface of the adsorbent during the ion-exchange process. In order to clarify this issue, we have extended discussion regarding the XRD and SEM result of ion-exchanged NMSC at the corresponding area. In addition, the wording ‘macropore formation’ from the previous draft seems to be inappropriate to describe the state of ion-exchanged NMSC. Since we cannot confirm whether these macropores were present before or newly formed after the ion-exchange, we replaced the previous wordings to the context of phase transition which seems to be the better description of the change. We have revised the manuscript as following:

- ✓ Structure variation upon the ion-exchange (L529-537)

“Disappearance of signature peaks (5°, 9.8°, 19.8°, 28.7°, 33°, and 50.5°) from Cs⁺-exchanged and Sr²⁺-exchanged samples suggest that ion-exchange process transformed the original layer structure of NMSC-1 into amorphous structure. In addition, the FE-SEM analysis of ion-exchanged NMSC-1 show the transition from layer structure to amorphous state in which this change aligns well with the XRD result. This ion-exchange-induced amorphous transition demonstrate the flexible nature of NMSC structure upon the exposure to the external stimuli (Supplementary Fig. 28). This ion-exchange-induced phase control is not the focus of this study,

but it would be an interesting theme of future study for potential connection to the field of reversible crystalline-to-amorphous phase transformation⁶⁴.”

[R1, Comments #7]

Some published papers closely related to the present work are suggested to be cited and commented such as Chem. Mater. 2013, 25, 2116–2127; Chem. Sci., 2016, 7, 1121–1132; J. Mater. Chem. A, 2016, 4, 16597–16605; Materials Lab, 2022 1, 220046.

[R1, Response #7]

Thank you for your kind comment. According to your suggestions, we added the references in the revised main manuscript.

✓ Revised references (L725-727, L730-731, L763-765, L784-786)

22. Sarma D., Islam S. M., Subrahmanyam K., Kanatzidis M. G. Efficient and selective heavy metal sequestration from water by using layered sulfide $K_{2x}Sn_{4-x}S_{8-x}$ ($x= 0.65-1$; KTS-3). J. Mater. Chem. A 4, 16597-16605 (2016).
24. Bai J., Yang L., Zhang Y., Sun X., Liu J. Tin Sulfide Chalcogel Derived SnS_x for CO_2 Electroreduction. Materials Lab 1, 1-4 (2022).
37. Sarma D., Malliakas C. D., Subrahmanyam K., Islam S. M., Kanatzidis M. G. $K_{2x}Sn_{4-x}S_{8-x}$ ($x= 0.65-1$): a new metal sulfide for rapid and selective removal of Cs^+ , Sr^{2+} and UO_2^{2+} ions. Chem. Sci. 7, 1121-1132 (2016).
44. Mertz J. L., Fard Z. H., Malliakas C. D., Manos M. J., Kanatzidis M. G. Selective Removal of Cs^+ , Sr^{2+} , and Ni^{2+} by $K_{2x}Mg_xSn_{3-x}S_6$ ($x= 0.5-1$) (KMS-2) Relevant to Nuclear Waste Remediation. Chem. Mater. 25, 2116-2127 (2013).

Response to the Reviewer 2's comment in the first review

The manuscript by Kim and co-authors reported a multiscale structural control method through the thiostannate motif-transformation-induced self-assembly process. The chalcogels fabricated in this work with a layer-by-layer stacking structure showed the adsorption performance of cesium and strontium ions. It is an interesting work, and the reported method may contribute to the design of unique chalcogels with different structural hierarchies. However, many important issues need to be addressed before the manuscript can be further considered for publication.

[R2, Comments #1]

The formation mechanism of the layered and amorphous structures is important for this work, and the illustration shown in Figures 2a-b could not be seen clearly. For readers' better understanding, I would recommend a reorganization of Figure 1 and Figure 2. Figures 2a-b are recommended to be enriched and placed in Figure 1.

[R2, Response #1]

Thank you for your kind comment. According to your suggestions, we combined and reform the Figure 1 and Figure 2a-b into Figure 1a-e. (Please see the Reviewer 1 Response #3 for Figure 1 reformation. L136-144) Now, new Figure 1 contain gelation process, EDS mapping, XRD, Raman, XPS, KMS-1 structure and SEM/mapping, and Mn:[Sn₂S₆]⁴⁻ ratio-induced gelation pattern distinction, providing an exclusive overview of the NMSC synthesis result. Former figure 2a-b is included as Figure 1e presenting Mn:[Sn₂S₆]⁴⁻ ratio-induced gelation pattern distinction of NMSC-1 (low Mn ratio) and NMSC-3 (high Mn ratio) which are representative example of slow and ordered or fast and disordered gelation, respectively.

✓ Figure 1 reformation (L136-144)

Fig. 1 Synthesis, characterization, and hypothesized mechanism of coordination (Tetrahedral-Octahedral)-structure (Layer-Amorphous) switching of sodium manganese sulfide chalcogel (NMSC) with a layered structure similar to that of metal sulfide (KMS-1) materials. **a** NMSC synthesis. **b** EDS mapping of layered NMSC-1, including Na, Mn, Sn,

and S. **c** Structural analyses of NMSC-1 using PXRD, FT-IR, and XPS analyses of NMSC-1 for Sn 3*d* and Mn 2*p* respectively. **d** Scheme and EDS mapping of layered KMS-1 structure, including K, Mn, Sn, and S. **e** Hypothesis mechanism of coordination (Tetrahedral-Octahedral) and structure (Layer-Amorphous) transformation in NMSC.

[R2, Comments #2]

Figure 5d showed the trend of the Cs⁺-adsorption capacity of NMSC-1 under various pH conditions. It seems that the best data might be in the range of pH = 4 ~ 6. However, the data at pH = 7 have the best performance of adsorption. Thus, the additional adsorption capacity data between pH = 6 and 8 could be good support.

[R2, Response #2]

This comment suggests the additional adsorption experiment on pH 6 and 8 to confirm the pH dependent behavior of NMSC-1. In addition to the reviewer's suggestion, we repeated ion-exchange experiment on all pH conditions to confirm the sorption pattern and the adsorption capacity of NMSC-1. We have updated pH dependent adsorption with its error bar and revised the manuscript at the corresponding areas as shown below.

✓ Figure 5 Revision (L457-465)

Fig. 5 Radionuclide remediation properties including error analysis of NMSCs. **a** Radionuclide (Cs^+ , Sr^{2+})-removal capacities of NMSCs in an aqueous solution at $\text{pH} = 7$. **b** Cesium adsorption capacities of the NMSCs at different manganese ratios, revealing the sorption-behavior dependence on the Na content over the total metal content in the structure. **c** Raw adsorption kinetic (Qt vs. time) of NMSC-1 for Cs^+ -exchanged (black line) and kinetic model of NMSC-1 described by pseudo-second-order fitting over the full equilibrium period (3 h) (blue line). **d** Cs^+ -adsorption capacity of NMSC-1 under various pH conditions. Error bars presents the standard deviation of the mean of three experiments.

✓ Ion-exchange standard deviation (L485-489)

“The error bars in Figure 5a represent the standard deviation of NMSC samples during Cs^+ and Sr^{2+} ion exchange. The standard deviation of NMSC-1 for Cs^+ removal capacity fluctuates in a broader range ($\pm 26 \text{ mg g}^{-1}$) due to the amounts of Na^+ loss in each wet gel synthesis. Other NMSCs illustrate a narrow standard deviation range ($\pm 7\text{-}11 \text{ mg g}^{-1}$), indicating the lesser ion-exchange activity.”

✓ Ion-exchange standard deviation II (L558-562)

“Adsorption stability at a wide range of pH is another unique feature of NMSC. At $\text{pH} 2\text{-}10$, NMSC-1 exhibited 37.4–56.1% ($\pm 4\text{-}26 \text{ mg g}^{-1}$ standard deviation) of the original Cs^+ -removal capacity (Fig. 5d), indicating a potential ion-exchange competition with H_3O^+ under acidic conditions and hydroxyl group attacks on the sulfidic sites under basic conditions⁶⁷.”

[R2, Comments #3]

What is the coordination structure of various linkers like? Do the chalcogels have a layered structure? The corresponding characterization should be supplemented.

[R2, Response #3]

Thank you for your kind comment. According to your suggestions, we described the analysis results of Na-Mg-Sn-S and Na-Sn(II)-Sn(IV)-S chalcogels in the main manuscript and added analysis data in supplementary information (SI). To confirm the coordination structure and

morphology of Na-Mg-Sn-S and Na-Sn(II)-Sn(IV)-S chalcogels, various analysis methods are applied such as Raman spectroscopy and FE-SEM. Other chalcogels (Na-Mg-Sn-S and Na-Sn(II)-Sn(IV)-S) showed the layer morphology in low stoichiometric Mg (0.5 : 1) or Sn(II) : Sn₂S₆ (0.25 : 1) ratio. In addition, the linker content increase tends to form the amorphous structure. The layer structure means octahedral coordination in metal center which is similar to the coordination structure of reported K_{1.38}Mg_{0.69}Sn_{1.31}S₄ (KMS-2) and K_{1.92}Sn_{3.04}S_{7.04} (KTS-3) materials. Furthermore, Raman spectrum of the other aerogels showed the significant signals (339 and 356 cm⁻¹ at 532 nm) of Sn-S octahedral vibration in tin cluster. Overall, other aerogel samples exhibited the crystal structure (Crystalline-Amorphous), morphology and coordination (Tetrahedral-Octahedral) transformation which is similar to the major NMSC materials. The updated results are shown below.

✓ Layer structure identification of NMgSC and NSn(II)SC (L419-L433)

“To further understand the coordination transformation and structure switching of NMgSC and NSn(II)SC aerogels, complimentary structure analyses of these chalcogels are conducted through Raman spectroscopy and FE-SEM (Supplementary Figs. 18-19). Raman spectrum of NMgSC and NSn(II)SC samples exhibit three prominent bands at 257, 339, and 356 cm⁻¹. The Sn₂S₂ ring vibration leads to a peak at 257 cm⁻¹. The peaks at 339 and 356 cm⁻¹ are ascribed to Sn-S octahedral vibration in the tin cluster (Supplementary Fig. 18)³⁷. The FE-SEM images reveal the morphological layer like flower flake for NMgSC-1 and NSn(II)SC-1 at low stoichiometric ratios. At high linker content, the layer morphology is damaged and switched to amorphous (NMgSC-2 and NSn(II)SC-2) (Supplementary Fig. 19).”

Supplementary Figure 18. Chemical bonding-nature of NMgSC and NSn(II)SC samples. Raman spectroscopy analysis of NMgSC and NSn(II)SC aerogels at different $[\text{Sn}_2\text{S}_6]^{4-}:\text{Mg}^{2+}/\text{Sn}^{2+}$ ratios. Peaks at 257, 339 and 356 cm^{-1} correspond to Sn_2S_2 ring and Sn-S octahedral vibrations at 532 nm.

Supplementary Figure 19. Morphological analysis of NMgSC and NSn(II)SC aerogel. FE-SEM results of (a-b) NMgSC-1 ($\text{Mg}^{2+}:\text{Sn}_2\text{S}_6 = 0.5:1$), (a-d) NMgSC-3 ($\text{Mg}^{2+}:\text{Sn}_2\text{S}_6 = 2:1$), (e-f) NSn(II)SC-1 ($\text{Sn}^{2+}:\text{Sn}_2\text{S}_6 = 0.25:1$) and (g-h) NSn(II)SC-3 ($\text{Sn}^{2+}:\text{Sn}_2\text{S}_6 = 2:1$).

[R2, Comments #4]

How is the stability of chalcogels during the adsorption process? Will the chalcogels swell? The and stability test for adsorption should be explored.

[R2, Response #4]

Thank you for your kind comment. According to your suggestions, we reveal the long-time running and stability test for Cs^+ adsorption of NMSC samples in the main manuscript and added analysis data in supplementary information (SI). After adding the solution, the dried gels get wet and breaks some species with volume shrinkage (Supplementary Figure 21b-c). During the consideration time (7 days), the NMSC samples are remained the physical shape (Supplementary Figure 21d).

✓ Chalcogels stability test (L493-498)

“To further understand the stability of chalcogels, the Cs^+ adsorption stability test is conducted. (Supplementary Fig. 21). The dried gel has a good wettability, so it is mechanically broken into smaller species in Cs^+ solution. During the equilibrium time (7 days), the physical shape of the gel is maintained. At the end of the stability test, chalcogel is converted to xerogel structure, leading to the pore collapse.”

Supplementary Figure 21. Stability of chalcogels during adsorption process in consideration time. (a) Dried gel chunks, (b) After adding 2 mL Cs^+ solution (100 ppm), (c) 12 h, and (d) 7 days.

[R2, Comments #5]

The authors should comment on why chose cesium and strontium for exploring the application potential of chalcogels? What are the advantages of chalcogels in this work compared to other cesium and strontium adsorption materials?

[R2, Response #5]

We have extended the discussion regarding the reason of choosing Cs⁺ and Sr²⁺ as potential adsorption target for NMSCs as following. Also, we added comparison table of other adsorption materials in the supplementary file to justify the application selection.

- ✓ Radionuclide capture motivation and result (L445-455)

“Cs⁺ and Sr²⁺ were selected as the targeting adsorbates due to their mid-range half-lives ($t_{1/2}$ = 30.2 and 28.8 years, respectively) and high fission yields (6.3 and 4.5%, respectively)⁵⁴. Precedent examples of Cs⁺ and Sr²⁺ capture materials exist in the form of inorganic compound^{29, 55, 56}, carbon heterostructure^{27, 57, 58}, Prussian blue pigments^{59, 60, 61}, metal-organic-framework⁶², and chalcogels¹³, but most of them are only functional in complex heterostructure form or require multiple steps of synthetic protocols (Supplementary Table 3). In contrast, NMSCs are anticipated to be functional for Cs⁺ and Sr²⁺ capture through ion-exchange principle, as precedent layered chalcogenide compound (e.g., KMS-1) are known to attract them with high efficiency. Without further functionalizations, the soft basic surface and Na-anchoring sites of NMSCs is anticipated to bound intermediate-to-soft cations of Cs⁺ and Sr²⁺, as other precedent chalcogenide aerogels have done successfully¹³.”

Supplementary Table 3. Cs⁺ and Sr²⁺ removal capacities of NMSC samples and selected high-performance materials.

Materials	Adsorbate	C _i ppm	Q _e mg·g ⁻¹	t (eq.) min	Ref.
KMS-1	Cs ⁺	1.1	226	60	1
CdSnSe-1	Cs ⁺	824	371.4	1440	8
	Sr ²⁺	741	128.4	1440	
PAN-KNiCF ^d	Cs ⁺	20~240	110.3	1440	9
GO-membrane on CaF ₂	Cs ⁺	87.3	148.0	30~1920	10
Graphene oxide	Cs ⁺	10	5.35	480	11
Na-GO fiber ^e	Cs ⁺	100	159.8	120 (10)	12
Copper hexacyanoferrate (CuHCF)	Cs ⁺	100-200	155.60	-	13
	Sr ²⁺	50-350	59.95	-	
PB-GO hydrogel beads ^a	Cs ⁺	665	164.5	600(600)	14
PSMGPB ^b	Cs ⁺	177300	213.9	1440(1440)	15
UiO66-NH-SO ₃ H-3 ^c	Cs ⁺	60	118.76	90	16
	Sr ²⁺	60	113.12	90	
TAC-3	Cs ⁺	100	191.1	180	4
NMSC-1	Cs ⁺	100	78 (average)	40 (180)	
	Sr ²⁺	100	41 (average)	(180)	
NMSC-2	Cs ⁺	100	40 (average)	-	This study
	Sr ²⁺	100	34 (average)	-	
NMSC-3	Cs ⁺	100	34 (average)	-	
	Sr ²⁺	100	23 (average)	-	

Q_e = equilibrium adsorption capacity. C_i = initial Cs⁺ concentration. t (eq.) = sorption time (equilibrium time)
^aPVA-alginate encapsulated PB-GO hydrogel beads. ^bpectin-stabilized magnetic graphene oxide Prussian blue nanocomposites ^cUiO-66-NH₂ with -SO₃H functional ^dPAN-based potassium nickel hexacyanoferrate (II) composite spheres. ^eNaOH 2wt%, 500 um, 20 mg/ml, thermal treatment 3h

✓ Revised supplementary references (SI, P41-42)

- Manos M. J., Kanatzidis M. G. Highly efficient and rapid Cs⁺ uptake by the layered metal sulfide K₂MnxSn_{3-x}S₆ (KMS-1). *J. Am. Chem. Soc.* 131, 6599-6607 (2009).
- Kang Y. K., et al. Thiostannate coordination transformation-induced self-crosslinking chalcogenide aerogel with local coordination control and effective Cs⁺ remediation functionality. *J. Mater. Chem. A* 8, 3468-3480 (2020).
- Zhu J.-Y., et al. Structural investigation of the efficient capture of Cs⁺ and Sr²⁺ by a microporous Cd-Sn-Se ion exchanger constructed from mono-lacunary supertetrahedral clusters. *Inorg. Chem. Front.*, (2022).
- Du Z., Jia M., Wang X. Cesium removal from solution using PAN-based potassium nickel hexacyanoferrate (II) composite spheres. *J. Radioanal. Nucl. Chem.* 298, 167-177 (2013).

10. Narayanam P. K., Jishnu A., Sankaran K. Graphene oxide supported filtration of cesium from aqueous systems. *Colloids Surf. A Physicochem. Eng. Asp.* 539, 416-423 (2018).
11. Xing M., Zhuang S., Wang J. Efficient removal of Cs (I) from aqueous solution using graphene oxide. *Prog. Nucl. Energy* 119, 103167 (2020).
12. Lee H., Lee K., Kim S. O., Lee J.-S., Oh Y. Effective and sustainable Cs⁺ remediation via exchangeable sodium-ion sites in graphene oxide fibers. *J. Mater. Chem. A* 7, 17754-17760 (2019).
13. Le L. H. T., et al. Prussian blue analogues of A₂[Fe(CN)₆] (A: Cu²⁺, Co²⁺, and Ni²⁺) and their composition-dependent sorption performances towards Cs⁺, Sr²⁺, and Co²⁺. *J. Nanomater.* 2021, (2021).
14. Jang J., Lee D. S. Enhanced adsorption of cesium on PVA-alginate encapsulated Prussian blue-graphene oxide hydrogel beads in a fixed-bed column system. *Bioresour. Technol.* 218, 294-300 (2016).
15. Kadam A. A., Jang J., Lee D. S. Facile synthesis of pectin-stabilized magnetic graphene oxide Prussian blue nanocomposites for selective cesium removal from aqueous solution. *Bioresour. Technol.* 216, 391-398 (2016).
16. Wu J., et al. Efficient removal of Sr²⁺ and Cs⁺ from aqueous solutions using a sulfonic acid-functionalized Zr-based metal-organic framework. *J. Radioanal. Nucl. Chem.* 328, 769-783 (2021).

[R2, Comment #6]

The adsorption of cesium and strontium in the current work cannot fully reflect the structural advantages of chalcogels. Can the chalcogels in this work capture cesium and strontium selectively? Do chalcogels have superior performance for the separation and adsorption of other ions, like uranium?

[R2, Response #6]

We appreciate the reviewer's valuable suggestion regarding the selectivity of chalcogels. As per the suggestion, the selective adsorption of Cs⁺ and Sr²⁺ would be important since these

radionuclides exist with other marine cations (e.g. Na^+) upon the exposure. Thus, we have conducted competitive adsorption experiment of Cs^+ over Na^+ and Sr^{2+} over Na^+ . Besides, we wanted to try the separation and adsorption experiments for uranium as your suggestion. However, the purchase of uranyl salt within South Korea is not permitted because of nuclear safety policy. We attached the response e-mail from the vendor (Merck) about the purchase of uranyl salt. We would be glad to try the uranyl salt adsorption if there is a legal way to conduct the experiment in our country or other legally permitted country by potential collaborators. The updated results are added in the following areas:

- ✓ The letter from the vendor about the purchase of uranyl salt (HT1011).

- ✓ Selectivity test result (L511-518)

“The competitive ion experiments were conducted to explore selective removal for Cs^+ or Sr^{2+} on the marine-cation existing condition, representatively, Na^+ (Supplementary Fig. 23 and Supplementary methods). The Cs^+ removal capacity of NMSC-1 in the dual Na^+ - Cs^+ solution decreased (from 4.7 to 11.3 mg g^{-1}) upon the Na^+ concentration increase, while there is no significant difference in the Sr^{2+} removal capacity (14.1 – 15.1 mg g^{-1}) in the dual Na^+ - Sr^{2+} solution. This trend demonstrated that divalent ion state (e.g. Sr^{2+}) is less susceptible to the competitive ion effect than monovalent ion state (e.g. Cs^+) in Na^+ -coexisting solution. This is another indirect evidence that NMSC control radionuclide through ion-exchange sorption.”

Supplementary Figure 23. Selective removal capacity of NMSC-1 in the Na⁺-Cs⁺ and Na⁺-Sr²⁺ co-existing solution at the different ratio. (Cs⁺ initial concentration : 10 ppm, Sr²⁺ initial concentration : 10 ppm).

[R2, Comment #7]

In the current manuscript, only the adsorption kinetic of chalcogels for cesium was discussed, the adsorption kinetic of chalcogels for strontium should also be added.

[R2, Response #7]

Thank you for your kind comment. According to your suggestions, we added the adsorption kinetic of strontium in supplementary information (SI). The adsorption kinetic of NMSC-1 for strontium shows the pseudo-second-order model ($R^2 \sim 0.999$, $Q_{e, cal} \sim 66.2$, $k \sim 0.059$) which is similar to the adsorption kinetic of NMSC-1 for cesium. The adsorption kinetic of NMSC-1 is shown below.

Supplementary Figure 22. Kinetic analysis of NMSC-1 for Sr²⁺ ion exchange. Raw adsorption kinetic (Q_t vs. time) of NMSC-1 for Sr²⁺-exchanged (black line) described by pseudo-second-order fitting over the full equilibrium period (3 h)

✓ Adsorption kinetics of Sr²⁺ (L500-501, L508-511)

“The adsorption behavior of NMSCs can be understood further through a kinetic analysis (Fig. 5c and Supplementary Fig. 22).”

“Based on this kinetic model fitting of the sorption data over a 180-minute period, NMSC-1 fits only the pseudo-second-order model ($R^2 \sim 0.991$, $Q_{e, cal} \sim 102$, $k \sim 0.00757$ (Cs⁺) and $R^2 \sim 0.999$, $Q_{e, cal} \sim 66.2$, $k \sim 0.0059$ (Sr²⁺)), indicating a chemisorption-oriented sorption behavior of the Na⁺-Cs⁺ and Na⁺-Sr²⁺ ion exchanges.”

[R2, Comment #8]

To help readers understand this field, the recent progress in the selective separation and adsorption of strontium and cesium ions from aqueous solutions should be mentioned in the introduction.

[R2, Response #8]

The current study's focus is exploring novel design of chalcogenide aerogel with well-known two dimensional structure motif as building blocks, we demonstrated radionuclide control as one of the potential application for this material. Since there are other prestigious examples of radionuclide capture materials with superior adsorption capacity or stability, we did not emphasize the recent trends of radionuclide capture field at the introduction. We thought adding recent trend of radionuclide capture at the introduction would distract the current theme of this study, so we added that requested information at the corresponding result section, instead.

✓ Recent radionuclide capture material results (L445-455)

“Cs⁺ and Sr²⁺ were selected as the targeting adsorbates due to their mid-range half-lives ($t_{1/2} = 30.2$ and 28.8 years, respectively) and high fission yields (6.3 and 4.5%, respectively)⁵⁴.”

Precedent examples of Cs^+ and Sr^{2+} capture materials exist in the form of inorganic compound^{29, 55, 56}, carbon heterostructure^{27, 57, 58}, Prussian blue pigments^{59, 60, 61}, metal-organic-framework⁶², and chalcogenides¹³, but most of them are only functional in complex heterostructure form or require multiple steps of synthetic protocols (Supplementary Table 3). In contrast, NMSCs are anticipated to be functional for Cs^+ and Sr^{2+} capture through ion-exchange principle, as precedent layered chalcogenide compound (e.g., KMS-1) are known to attract them with high efficiency. Without further functionalizations, the soft basic surface and Na-anchoring sites of NMSCs is anticipated to bound intermediate-to-soft cations of Cs^+ and Sr^{2+} , as other precedent chalcogenide aerogels have done successfully¹³.”

[R2, Comment #9]

The characterization (e.g., surface charge) for NMSC should be added and further discussed. Optical images of the aerogel and its wet-gel state should be modified because they cannot reflect the similarity of state and volume mentioned in the article.

[R2, Response #9]

We appreciate the reviewer’s valuable suggestion regarding the additional characterization and optical images of the NMSC aerogel. We performed the isoelectric point (point of zero charge, PZC) for all NMSC samples by the salt addition method (Supplementary Figure 30). The ΔpH was plotted against the initial pH values, and the isoelectric point is the pH at which ΔpH is zero. In the pH 2-6 range, the ΔpH values are positive with the maximum value at pH 3 (NMSC-1 and NMSC-2) and pH 4 (NMSC-3). The isoelectric points of NMSCs are 7.13 (NMSC-1), 7.2 (NMSC-2) and 6.5 (NMSC-3) at 298 K. According to the plot, the radionuclides adsorption (Cs^+ and Sr^{2+}) of NMSC should be better at the base solution by physical adsorption support of negative charge on the surface. However, the removal capacity result of NMSC-1 showed its highest capacity obtained at neutral charge surface. Therefore, we can conclude that the adsorption mechanism of NMSC depends on the chemical adsorption by ion-exchange mechanism only. In addition, the optical images of NMSC samples show that the physical shape stability of wet gel is directly proportional to the added amount of manganese. In addition, the physical shape of wet gel is maintained after critical point drying process (Supplementary Figure 2).

- ✓ Optical images of wet gel and CPD aerogel of NMSCs (Supplementary Information)

Supplementary Figure 2. Optical images of (a, e) NMSC-1, (b, f) NMSC-2, and (c, g) NMSC-3 before and after critical point drying (CPD).

Supplementary Figure 30. The ΔpH versus initial pH of NMSC suspensions at 298 K.

- ✓ Physical NMSC aerogel appearance (L123-125)

“With subsequent solvent-exchange and critical point drying (CPD) processes, a monolithic and homogeneous light-colored aerogel was produced, retaining the volume of its wet-gel state (Fig. 1a and Supplementary Figs. 1-2).”

- ✓ Surface charge analysis (L548-558)

“The influence of the charge on the NMSC surface during the adsorption is measured by the isoelectric point (point of zero charge, PZC). It is analyzed by the salt addition method⁶⁶. The

ΔpH was plotted against the initial pH values, and the isoelectric point is the pH at which ΔpH is zero (Supplementary Fig. 30). In the pH 2-6 range, the ΔpH values are positive with the maximum value at pH 3 (NMSC-1 and NMSC-2) and pH 4 (NMSC-3). The isoelectric points of NMSC are 7.13 (NMSC-1), 7.2 (NMSC-2), and 6.5 (NMSC-3) at 298 K, respectively. Based on the graph, the radionuclides adsorption (Cs^+ and Sr^{2+}) of NMSC should be better at the base solution by physical adsorption support of negative charge on the surface. However, the removal capacity of NMSC-1 showed its highest capacity obtained at a neutral charge surface. Therefore, we can conclude that chemisorption-induced ion-exchange is the dominant radionuclide remediation principle.”

[R2, Comment #10]

The snapshots for water contact angle on the surface of chalcogels should be added. In this work, the authors proposed that chalcogels can be used for aqueous radionuclide-adsorption, and the surface wettability is important for the adsorption ability.

[R2, Response #10]

Thank you for your suggestion about the wettability of our materials. We measured the water contact angle of surface of NMSC-1 as the representative among NMSCs. NMSC-1 showed it has very high water wettability on the surface and it means NMSC-1 can be an appropriate adsorbent on the aqueous radionuclide waste solution. The revised manuscript and added supplementary figure are as shown below.

- ✓ Water Contact-angle measurement of NMSC-1 (Supplementary Information)

Supplementary Figure 24. Water Contact-angle measurement of NMSC-1.

✓ Water contact angle result (L519-523)

“Meanwhile, the unique morphology of NMSC-1 with layered structure can be examined using surface wettability measurement. The small water contact angle of NMSC-1 ($\theta \sim 11^\circ$) shows its hydrophilic surface with high surface energy. The high water wettability of NMSC-1 proposes the potential for candidate of Cs^+ and Sr^{2+} adsorbent on the aqueous solution state (Supplementary Fig. 24 and Supplementary method).”

[R2, Comment #11]

In general, the error analysis of the chalcogels should be added, especially in the experiments on the adsorption capacity of radial ions. Also, the necessary labels of the y-axis should be complemented.

[R2, Response #11]

Thank you for the feedback related to error analysis of chalcogels. We repeated the adsorption experiments (Cs^+ and Sr^{2+}) and pH conditions in range of 2-10 (Cs^+) for calculating the standard deviation (error bars). The revised Fig. 5 is redraw as shown below.

✓ Revised Fig. 5 (L457-465)

Fig. 5 Radionuclide remediation properties including error analysis of NMSCs. a Radionuclide (Cs^+ , Sr^{2+})-removal capacities of NMSCs in an aqueous solution at pH = 7. **b**

Cesium adsorption capacities of the NMSCs at different manganese ratios, revealing the sorption-behavior dependence on the Na content over the total metal content in the structure. **c** Raw adsorption kinetic (Q_t vs. time) of NMSC-1 for Cs^+ -exchanged (black line) and kinetic model of NMSC-1 described by pseudo-second-order fitting over the full equilibrium period (3 h) (blue line). **d** Cs^+ -adsorption capacity of NMSC-1 under various pH conditions. Error bars presents the standard deviation of the mean of three experiments.

✓ Standard deviation addition (L485-489)

“The error bars in Figure 5a represent the standard deviation of NMSC samples during Cs^+ and Sr^{2+} ion exchange. The standard deviation of NMSC-1 for Cs^+ removal capacity fluctuates in a broader range ($\pm 26 \text{ mg g}^{-1}$) due to the amounts of Na^+ loss in each wet gel synthesis. Other NMSCs illustrate a narrow standard deviation range ($\pm 7-11 \text{ mg g}^{-1}$), indicating the lesser ion-exchange activity.”

✓ Standard deviation addition II (L559-562)

“At pH 2–10, NMSC-1 exhibited 37.4–56.1% ($\pm 4-26 \text{ mg g}^{-1}$ standard deviation) of the original Cs^+ -removal capacity (Fig. 5d), indicating a potential ion-exchange competition with H_3O^+ under acidic conditions and hydroxyl group attacks on the sulfidic sites under basic conditions⁶⁶. ”

Response to the Reviewer 3's comment in the first review

The manuscript titled “Multiscale structural control of thiostannate chalcogels with two dimensional crystalline constituents” presents the synthesis, structural characterization, and ion exchange properties of a new family of Mn based chalcogenide aerogels exhibiting KMS-1 like layered elements. The authors claim that the porosity, macrostructure, crystallinity, and kinetics are sensitive to the Sn:Mn ratio in the precursor solutions. The authors go on to support these claims by preparing three distinct materials utilizing this method of control; they are then extensively studied via an array of spectroscopic and scattering techniques. Their findings are important and of broad interest to the materials chemistry, material science, and ion exchange communities. A major finding of this work is the control of structure which may enable a hierarchical structural design of future chalcogel materials that has been limited by existing synthetic techniques. While the ion exchange characterization of these materials is enough to imply their ability to ion exchange, more studies are needed to understand their ion exchange performance fully. Without both a detailed study on the ion exchange capabilities of these materials and evidence that the synthetic control they have developed can be used to enhance this capability, it is difficult to justify the acceptance of this manuscript to this journal. Overall, this manuscript presents an important and potentially impactful contribution to the existing literature on chalcogenide aerogels and is recommended for publication in the Journal Nature Communications following the major revisions detailed below. Prior to publication, the authors should address the following major issues.

[R3, Comment #1]

Figure 2a-b is somewhat misleading to the reader due to the depicted coordination environment of Mn (square planer to my eye). The authors do not comment on the coordination of Mn in the text beyond correctly stating that it prefers octahedral environments. This figure thus implies a mechanistic understanding of the assembly which is not supported by the data presented. It would be more appropriate to depict the interaction of Mn with the $[\text{Sn}_2\text{S}_4]^{4-}$ cluster using exclusively arrows do communicate only chelation and not coordination. In addition, because there is no real mechanistic data both the t-IMTC unit and the image depicting the transformation should be removed/altered. Such a reaction scheme should be constructed

using pictures, not bonds, to avoid implying a local understanding of this process. The image below shows the interactions I'm referring to circled in red along with an example of how the chelation interaction could be drawn.

[R3, Response #1]

Thank you for your kind comment. According to your suggestions, we redrew Figure 1 including Figure 2a-b and added a scheme of KMS-1 material. Now, Fig. 1 contain gelation process, EDS mapping, XRD, Raman, XPS, KMS-1 structure and SEM/mapping, and Mn:[Sn₂S₆]⁴⁻ ratio-induced gelation pattern distinction, providing an exclusive overview of the NMSC synthesis result. Furthermore, we rewrote the gelation mechanism and confirmed the t-IMTC unit which engages in the mechanism by Raman spectroscopy of [Sn₂S₆] cluster and NMSC samples. The revised Figure 1 is shown below. In Figure 1d, more detailed structural description of KMS-1 is added to confirm the structural similarity with NMSCs. Figure 1e presents Mn:[Sn₂S₆]⁴⁻ ratio-induced gelation pattern distinction of NMSC-1 (low Mn ratio) and NMSC-3 (high Mn ratio) which are representative example of slow and ordered or fast and disordered gelation, respectively.

✓ Revised Fig. 1 (L136-144)

Fig. 1 Synthesis, characterization, and hypothesized mechanism of coordination (Tetrahedral-Octahedral)-structure (Layer-Amorphous) switching of sodium manganese sulfide chalcogel (NMSC) with a layered structure similar to that of metal sulfide (KMS-

1) materials. a NMSC synthesis. **b** EDS mapping of layered NMSC-1, including Na, Mn, Sn, and S. **c** Structural analyses of NMSC-1 using PXRD, FT-IR, and XPS analyses of NMSC-1 for Sn *3d* and Mn *2p* respectively. **d** Scheme and EDS mapping of layered KMS-1 structure, including K, Mn, Sn, and S. **e** Hypothetical mechanism of coordination (Tetrahedral-Octahedral) and structure (Layer-Amorphous) transformation in NMSC.

✓ Raman spectroscopy of NMSC (Supplementary Information)

Supplementary Figure 9. Chemical bonding-nature analysis of NMSC. Raman spectroscopy analysis of Na₄Sn₂S₆ and NMSC at different [Sn₂S₆]⁴⁻:Mn²⁺ ratios at wavelength (a) 785 and (b) 532 nm.

✓ Raman analysis of NMSC (L201-217)

“Moreover, the Raman spectrum of Na₄Sn₂S₆ demonstrated the structure of [Sn₂S₆]⁴⁻ cluster (Supplementary Fig. 9a). The peak at 385 cm⁻¹ is responsible for the resonance of symmetric Sn-S_{terminal} stretching mode, and the tetrahedral Sn-S vibration in the bridging site presents at 346 cm⁻¹. The peak values in a range of between 346 and 385 cm⁻¹ are assigned to the isolated [Sn₂S₆]⁴⁻ anions. The Sn-S_{bridge} vibration occurs at 328 cm⁻¹. The Sn₂S₂ ring vibration is located at 287 cm⁻¹. The peaks below 200 cm⁻¹ are responsible for deformation vibration of SnS₂ wagging and twisting modes³⁶. However, after coordinated with Mn²⁺, NMSC-1 displays prominent vibrational peaks in the 271, 314, and 356 cm⁻¹ in Raman spectroscopy (Supplementary Fig. 9a), indicating Sn₂S₂ ring vibrations and Sn-S vibrations in the octahedral tin cluster^{13, 37}. The disappearance of Sn-S_{terminal} vibration and retention of the Sn₂S₂ ring vibration indicate self-assembly between the terminal sulfur of [Sn₂S₆]⁴⁻ cluster and Mn²⁺ linker. In the NMSC-2 and NMSC-3, a peak at 348 cm⁻¹ presents tetrahedral Sn-S vibration in

bridging site at marginally higher energy, corresponding to the presence of a tetrahedral tin center (Supplementary Fig. 9b)³⁸. In addition, the peaks at 218 and 472 cm^{-1} suggest Mn–S vibrations and sodium–sulfide interactions (Supplementary Fig. 9b)³⁹. This indirectly proves the tetrahedral-to-octahedral transformation of the thiostannate cluster during the sol–gel self-assembly reaction.”

✓ KMS-1 material characterizations (L246-253)

“To confirm the two-dimensional local structural motif of NMSC, KMS-1, the mother structure of NMSCs, is synthesized and characterized through SEM, and EDS analysis. (Fig. 1d). The typical metal layer sulfides (KMSn_2S_6) families) in the space group of $R\bar{3}m$ includes KMS-1 ($\text{M} = \text{Mn}^{2+}$)²⁹, KMS-2 ($\text{M} = \text{Mg}^{2+}$)⁴⁴, and KMS-5 ($\text{M} = \text{In}^{3+}$)⁴⁵. The layer is constructed by (M/Sn) S_6 octahedral coordination with M and Sn atoms occupying the same position. These structures are generally obtained by high temperature and pressure hydrothermal synthesis. We anticipate that NMSCs develop the structural motif of KMS-1 with the Na^+ replacement at the K^+ position during the spontaneous gelation.”

✓ Stoichiometric variation-induced structure formation (L256-271)

“All three NMSCs present a common metathesis reaction with a donor-acceptor bond between Mn^{2+} and tetrahedral thiostannate cluster ($\text{t-}[\text{Sn}_2\text{S}_6]^{4+}$). If slow gelation allows thermodynamically stable phase formation, as in the case of NMSC-1 formation, the manganese center favors a densely packed octahedral coordination and causes a shift in the core coordination of the Sn^{4+} center from tetrahedral to octahedral. Resulting octahedral coordinated manganese thiostannate cluster triggers layer-by-layer stacking aggregation to form the layered NMSC-1 structure, as shown in Fig. 1e (1). However, increasing the molar ratio of $[\text{Sn}_2\text{S}_6]:\text{Mn}$ to 1:1 (NMSC-2) or 1:2 (NMSC-3) stimulates the gelation process, leading to a mixed tetrahedral-octahedral coordination formation of the Sn^{4+} center. The abundant stoichiometric metal linking centers (Mn^{2+}) facilitated a rapid assembly of mixed Sn^{4+} coordination. As a result, the mixed coordination of Sn^{4+} center creates a three-dimensional amorphous macrostructure with short-range ordering and random orientation. This random self-cross-linking behavior becomes more dominant as the stoichiometric Mn^{2+} ratio increase, as in the case of NMSC-3 (Fig. 1e (2)). Thus, we speculate that the stoichiometric- Mn^{2+} -

variations play a crucial role in determining the coordination mode of NMSC and its corresponding porous structure.”

[R3, Comment #2]

On line 272 it states “The BET surface areas of NMSC-1, NMSC-2, and NMSC-3 increase with the Mn content in $[\text{Sn}_2\text{S}_6]:\text{Mn}$ to 115, 115, and 250 $\text{m}^2 \text{g}^{-1}$ ($R^2 \sim 0.999$), respectively.” Why do NMSC-1 and NMSC-2 have the same surface areas, given their differences in structure? Shouldn't the surface area of NMSC-2 > NMSC-1?

[R3, Response #2]

Thank you for the feedback related to the BET surface statement. As reviewer raise the issue, the previous result did not confirm the strong correlation between BET surface area and stoichiometric Mn content. Thus, we conducted additional BET surface area to examine the reproducibility and stability of NMSCs' porous structure. In the previous manuscript, only the representative BET surface areas with the highest R^2 value is reported in the text. To raise the reliability of result, the BET surface area of NMSCs is now reported with the average values (Supplementary Table 2). The contents of Figure 2 (N_2 isotherms of NMSC-1 and NMSC-3) remains same because these are the record high surface area values for NMSC-1 and NMSC-3, respectively. Considering the random aggregation environment of NMSCs gelation, we decided to explain the consequences of controlling $\text{Mn}:[\text{Sn}_2\text{S}_6]^{4-}$ ratio in the porous structure formation instead of giving an affirmative trend of Mn content and BET surface area. Since, the newly obtained NMSC-2 sample's isotherm and BJH result break the previous record, the representative figures of NMSC-2 are replaced to new one. (Supplementary Information) Changes are made as following:

✓ Mn^{2+} variation impact on NMSCs porous structure formation (L25-29)
“The aerogels exhibited $\text{Mn}:[\text{Sn}_2\text{S}_6]^{4-}$ stoichiometric-variation-induced-control of average specific surface areas (95–226 $\text{m}^2 \text{g}^{-1}$), thiostannate coordination networks (octahedral to tetrahedral), phase crystallinity (crystalline to amorphous), and hierarchical porous structures (micropore-intensive to mixed-pore state).”

✓ Mn^{2+} variation impact on NMSCs porous structure formation (L296-303)

“The BET average surface areas of NMSC-1, NMSC-2, and NMSC-3 is 95, 124, and 226 m² g⁻¹ ($R^2 \sim 0.999$), respectively (Supplementary Table 2). By controlling Mn²⁺:[Sn₂S₆]⁴⁻ ratio, the porous structure formation pattern varies. We speculate that large surface area difference between NMSC-1 and NMSC-3 is related to the production of amorphous domains within the gel network where meso- and macropores are more likely forming via random aggregation. This maybe the probable reason that NMSC-3, in which contains the most degree of amorphous domains, possess the largest BET surface area of all NMSCs. Large discrepancies within the surface area of each NMSCs explain the random aggregation nature sol-gel reaction. ”

Supplementary Table 2. The BET Surface area values of NMSCs.

Samples	BET surface area (m ² g ⁻¹)			Average
NMSC-1	73	96	115	95
NMSC-2	101	114	156	124
NMSC-3	200	229	250	226

- ✓ The isotherm and BJH pore size distribution of NMSC-2 (L306-308)

“Both NMSC-1 and NMSC-2 exhibit well-developed meso-porosities, with interparticle pores characterized by pore diameters of 3.72-3.81 nm on the NMSC layer, whereas NMSC-3 exhibits no specific mesopores⁴⁸ ”

Supplementary Figure 14. Pore-structure analysis of NMSC-2. Brunauer–Emmett–Teller (BET) measures for (a) specific surface area and (b) pore size distribution of NMSC-2.

[R3, Comment #3]

The BET surface areas should be measured both before and after ion exchange to observe changes in the surface area that come from this process.

[R3, Response #3]

As the reviewer suggest, we have conducted N₂ isotherm of NMSC-1 after the ion-exchange. Corresponding changes are made as following:

- ✓ BET Surface area of NMSC-1 after the ion-exchange process. (Supplementary Information)

Supplementary Figure 28. Pore-structure analysis of NMSC-1 after Cs⁺ and Sr²⁺ ion exchanges. Brunauer–Emmett–Teller (BET) measures of NMSC-1 for (a) Cs-exchanged (b) Sr-exchanged.

- ✓ Surface area measurement after the ion-exchange (L541-545)

“Furthermore, the surface area of NMSC-1 after Cs⁺ ion exchange is measured by BET observing the changes on the surface area before and after ion exchange. The BET results show the collapsed surface area of NMSC-1 after Cs⁺ (2.3 m² g⁻¹) and Sr²⁺ (3.3 m² g⁻¹) ion exchanges (Supplementary Fig. 28). This demonstrates irreversible nature of ion-exchange sorption of NMSCs.”

[R3, Comment #4]

EDS-SEM analysis should also be conducted before and after the exchange to confirm loading of the Cs/Sr in the materials and the absence/reduction of Na in the compounds. Figure S17 depicts the SEM micrographs of the materials post ion exchange but does not show the EDS of the materials as well.

[R3, Response #4]

Thank you for your kind comment. According to your suggestions, we explained the results in the main manuscript and added the data in supplementary information. We added the EDS mapping images and Element & Atomic % table of NMSC-1 after Cs⁺ and Sr²⁺ adsorption which showed uniformly the elemental contribution of the Cs/Sr in the materials and the absence/reduction of Na in the NMSC-1 material.

- ✓ Mapping result of ion-exchanged NMSC-1 (Supplementary Information)

Supplementary Figure 27. EDS mapping of NMSC-1 after ion exchanges. (a) Cs⁺ and (b) Sr²⁺ include Cs, Sr, Mn, Sn, and S elements.

Supplementary Table 4. Element & Atomic % of NMSC-1 after Cs⁺ and Sr²⁺ adsorption with absence/reduction of Na.

Element	Element & Atomic %					
	Na	Cs	Sr	Mn	Sn	S
Cs-exchanged	1.73	10.31	-	13.58	30.21	44.17
Sr-exchanged	0.61	-	3.20	12.44	36.77	46.99

✓ EDS mapping analysis of ion-exchanged NMSC-1 (L538-541)

“In addition, EDS mapping images and atomic % (Cs, Sr, Mn, Sn, and S) of NMSC-1 after Cs⁺ and Sr²⁺ adsorption exhibit the complete replacement of Na⁺ with Cs⁺ or Sr²⁺ after the ion exchange (Supplementary Fig. 27 and Supplementary Table 4), confirming the ion-exchange mechanism.”

[R3, Comment #5]

The authors should include a comparison of the performance of their materials with other Cs sorbent materials present in the literature to better contextualize their results.

[R3, Response #5]

As the reviewer suggested, we have added comparative Cs sorption materials in the table form as following:

Supplementary Table 3. Cs⁺ and Sr²⁺ removal capacities of NMSC samples and selected high-performance materials.

Materials	Adsorbate	C _i ppm	Q _e mg·g ⁻¹	t (eq.) min	Ref.
KMS-1	Cs ⁺	1.1	226	60	1
CdSnSe-1	Cs ⁺	824	371.4	1440	8
	Sr ²⁺	741	128.4	1440	
PAN-KNiCF ^d	Cs ⁺	20~240	110.3	1440	9
GO-membrane on CaF ₂	Cs ⁺	87.3	148.0	30~1920	10
Graphene oxide	Cs ⁺	10	5.35	480	11
Na-GO fiber ^e	Cs ⁺	100	159.8	120 (10)	12
Copper hexacyanoferrate (CuHCF)	Cs ⁺	100-200	155.60	-	13
	Sr ²⁺	50-350	59.95	-	
PB-GO hydrogel beads ^a	Cs ⁺	665	164.5	600(600)	14
PSMGPB ^b	Cs ⁺	177300	213.9	1440(1440)	15
UiO66-NH-SO ₃ H-3 ^c	Cs ⁺	60	118.76	90	16
	Sr ²⁺	60	113.12	90	
TAC-3	Cs ⁺	100	191.1	180	4
NMSC-1	Cs ⁺	100	78 (average)	40 (180)	
	Sr ²⁺	100	41 (average)	(180)	
NMSC-2	Cs ⁺	100	40 (average)	-	This study
	Sr ²⁺	100	34 (average)	-	
NMSC-3	Cs ⁺	100	34 (average)	-	
	Sr ²⁺	100	23 (average)	-	

Q_e = equilibrium adsorption capacity. C_i = initial Cs⁺ concentration. t (eq.) = sorption time (equilibrium time)
^aPVA-alginate encapsulated PB-GO hydrogel beads. ^bpectin-stabilized magnetic graphene oxide Prussian blue nanocomposites ^cUiO-66-NH₂ with -SO₃H functional ^dPAN-based potassium nickel hexacyanoferrate (II) composite spheres. ^eNaOH 2wt%, 500 um, 20 mg/ml, thermal treatment 3h

✓ Revised supplementary references (SI, P41-42)

- Manos M. J., Kanatzidis M. G. Highly efficient and rapid Cs⁺ uptake by the layered metal sulfide K_{2x}Mn_xSn_{3-x}S₆ (KMS-1). *J. Am. Chem. Soc.* 131, 6599-6607 (2009).
- Kang Y. K., et al. Thiostannate coordination transformation-induced self-crosslinking chalcogenide aerogel with local coordination control and effective Cs⁺ remediation functionality. *J. Mater. Chem. A* 8, 3468-3480 (2020).
- Zhu J.-Y., et al. Structural investigation of the efficient capture of Cs⁺ and Sr²⁺ by a microporous Cd-Sn-Se ion exchanger constructed from mono-lacunary supertetrahedral clusters. *Inorg. Chem. Front.*, (2022).
- Du Z., Jia M., Wang X. Cesium removal from solution using PAN-based potassium nickel hexacyanoferrate (II) composite spheres. *J. Radioanal. Nucl. Chem.* 298, 167-177 (2013).

10. Narayanam P. K., Jishnu A., Sankaran K. Graphene oxide supported filtration of cesium from aqueous systems. *Colloids Surf. A Physicochem. Eng. Asp.* 539, 416-423 (2018).
11. Xing M., Zhuang S., Wang J. Efficient removal of Cs (I) from aqueous solution using graphene oxide. *Prog. Nucl. Energy* 119, 103167 (2020).
12. Lee H., Lee K., Kim S. O., Lee J.-S., Oh Y. Effective and sustainable Cs⁺ remediation via exchangeable sodium-ion sites in graphene oxide fibers. *J. Mater. Chem. A* 7, 17754-17760 (2019).
13. Le L. H. T., et al. Prussian blue analogues of A₂[Fe(CN)₆] (A: Cu²⁺, Co²⁺, and Ni²⁺) and their composition-dependent sorption performances towards Cs⁺, Sr²⁺, and Co²⁺. *J. Nanomater.* 2021, (2021).
14. Jang J., Lee D. S. Enhanced adsorption of cesium on PVA-alginate encapsulated Prussian blue-graphene oxide hydrogel beads in a fixed-bed column system. *Bioresour. Technol.* 218, 294-300 (2016).
15. Kadam A. A., Jang J., Lee D. S. Facile synthesis of pectin-stabilized magnetic graphene oxide Prussian blue nanocomposites for selective cesium removal from aqueous solution. *Bioresour. Technol.* 216, 391-398 (2016).
16. Wu J., et al. Efficient removal of Sr²⁺ and Cs⁺ from aqueous solutions using a sulfonic acid-functionalized Zr-based metal-organic framework. *J. Radioanal. Nucl. Chem.* 328, 769-783 (2021).

- ✓ Radionuclide capture motivation and result of NMSCs in comparison to the benchmark materials (L445-455)

“Cs⁺ and Sr²⁺ were selected as the targeting adsorbates due to their mid-range half-lives ($t_{1/2}$ = 30.2 and 28.8 years, respectively) and high fission yields (6.3 and 4.5%, respectively)⁵⁴. Precedent examples of Cs⁺ and Sr²⁺ capture materials exist in the form of inorganic compound^{29, 55, 56}, carbon heterostructure^{27, 57, 58}, Prussian blue pigments^{59, 60, 61}, metal-organic-framework⁶², and chalcogels¹³, but most of them are only functional in complex heterostructure form or require multiple steps of synthetic protocols (Supplementary Table 3). In contrast, NMSCs are anticipated to be functional for Cs⁺ and Sr²⁺ capture through ion-exchange principle, as

precedent layered chalcogenide compound (e.g., KMS-1) are known to attract them with high efficiency. Without further functionalizations, the soft basic surface and Na-anchoring sites of NMSCs is anticipated to bound intermediate-to-soft cations of Cs^+ and Sr^{2+} , as other precedent chalcogenide aerogels have done successfully¹³.”

✓ Revised references (L699-701, L737-739, L743-744, L810-833)

13. Kang Y. K., *et al.* Thiostannate coordination transformation-induced self-crosslinking chalcogenide aerogel with local coordination control and effective Cs^+ remediation functionality. *J. Mater. Chem. A* **8**, 3468-3480 (2020).
27. Lee H., Lee K., Kim S. O., Lee J.-S., Oh Y. Effective and sustainable Cs^+ remediation via exchangeable sodium-ion sites in graphene oxide fibers. *J. Mater. Chem. A* **7**, 17754-17760 (2019).
29. Manos M. J., Kanatzidis M. G. Highly efficient and rapid Cs^+ uptake by the layered metal sulfide $\text{K}_{2x}\text{Mn}_x\text{Sn}_{3-x}\text{S}_6$ (KMS-1). *J. Am. Chem. Soc.* **131**, 6599-6607 (2009).
54. Jin K., Lee B., Park J. Metal-organic frameworks as a versatile platform for radionuclide management. *Coord. Chem. Rev.* **427**, 213473 (2021).
55. Zhu J.-Y., *et al.* Structural investigation of the efficient capture of Cs^+ and Sr^{2+} by a microporous Cd–Sn–Se ion exchanger constructed from mono-lacunary supertetrahedral clusters. *Inorg. Chem. Front.*, (2022).
56. Du Z., Jia M., Wang X. Cesium removal from solution using PAN-based potassium nickel hexacyanoferrate (II) composite spheres. *J. Radioanal. Nucl. Chem.* **298**, 167-177 (2013).
57. Xing M., Zhuang S., Wang J. Efficient removal of Cs (I) from aqueous solution using graphene oxide. *Prog. Nucl. Energy* **119**, 103167 (2020).
58. Narayanam P. K., Jishnu A., Sankaran K. Graphene oxide supported filtration of cesium from aqueous systems. *Colloids Surf. A Physicochem. Eng. Asp.* **539**, 416-423 (2018).
59. Le L. H. T., *et al.* Prussian blue analogues of $\text{A}_2[\text{Fe}(\text{CN})_6]$ (A: Cu^{2+} , Co^{2+} , and Ni^{2+}) and their composition-dependent sorption performances towards Cs^+ , Sr^{2+} , and Co^{2+} . *J. Nanomater.* **2021**, (2021).

60. Jang J., Lee D. S. Enhanced adsorption of cesium on PVA-alginate encapsulated Prussian blue-graphene oxide hydrogel beads in a fixed-bed column system. *Bioresour. Technol.* **218**, 294-300 (2016).
61. Kadam A. A., Jang J., Lee D. S. Facile synthesis of pectin-stabilized magnetic graphene oxide Prussian blue nanocomposites for selective cesium removal from aqueous solution. *Bioresour. Technol.* **216**, 391-398 (2016).
62. Wu J., *et al.* Efficient removal of Sr²⁺ and Cs⁺ from aqueous solutions using a sulfonic acid-functionalized Zr-based metal-organic framework. *J. Radioanal. Nucl. Chem.* **328**, 769-783 (2021).

[R3, Comment #6]

Competitive ion exchange studies should be conducted with the materials to understand their selectivity and enable comparison with existing materials.

[R3, Response #6]

As suggested, we have conducted competitive Cs⁺ and Sr²⁺ adsorption test against other common marine cations (e.g. Na⁺) and updated at the adsorption result section.

- ✓ Competitive ion-exchange result (L511-518)

“The competitive ion experiments were conducted to explore selective removal for Cs⁺ or Sr²⁺ on the marine-cation existing condition, representatively, Na⁺ (Supplementary Fig. 23 and Supplementary methods). The Cs⁺ removal capacity of NMSC-1 in the dual Na⁺ - Cs⁺ solution decreased (from 4.7 to 11.3 mg g⁻¹) upon the Na⁺ concentration increase, while there is no significant difference in the Sr²⁺ removal capacity (14.1 – 15.1 mg g⁻¹) in the dual Na⁺ - Sr²⁺ solution. This trend demonstrated that divalent ion state (e.g. Sr²⁺) is less susceptible to the competitive ion effect than monovalent ion state (e.g. Cs⁺) in Na⁺-coexisting solution. This is another indirect evidence that NMSC control radionuclide through ion-exchange sorption.”

Supplementary Figure 23. Selective removal capacity of NMSC-1 in the Na⁺-Cs⁺ and Na⁺-Sr²⁺ co-existing solution at the different ratio. (Cs⁺ initial concentration: 10 ppm, Sr²⁺ initial concentration : 10 ppm).

[R3, Comment]

In addition to the above Major revisions, the authors should also address the following minor revisions:

[R3, Response]

Reviewer's comment #6-1 ~ 6-5 are renumbered as #7 - 11

[R3, Comment #7]

The title should be changed to communicate the ion exchange experiments done with these materials.

[R3, Comment #7]

Thank you for the constructive suggestion regarding the ion-exchange element addition on the title. We understand that this suggestion is very logical from the reviewer's view point, but we would like to share the reason that we have excluded ion-exchange keyword from the title at the first manuscript. Through this study, we wanted to show the novel example of three dimensional chalcogen which is composed of KMS-1 like well-defined two dimensional

structural motif. Although we chose ion-exchange as potential application, ion-exchange result wasn't the focus of our study from the beginning. In addition, there are other precedent adsorbents showing the superior performance of radionuclide remediation. So we thought that emphasizing the 'ion-exchange' keyword on the title would provide the inappropriate impression to the scope of the current study. With this rationale, we hope that our decision on excluding ion-exchange keyword is convincing enough. If the majority of review committee members still think our material's ion-exchange functionality is of full merit in emphasizing, we would be glad accepting such a suggestion after all.

[R3, Comment #8]

Line 55 "adamantine clusters ($[M_4S_{10}]^{4-}$, M = Sn, Ge/Q = S, Se)" I think it is Q instead of S.

[R3, Response #8]

Thank you for the pointing out the typo. We have corrected as suggested.

✓ **Typo correction (L53)**

"...adamantine clusters ($[M_4Q_{10}]^{4-}$..."

[R3, Comment #9]

On Lines 32, 105, 407, 479. The word "selective" should be removed. It is not appropriate to describe these materials as selective as competitive experiments (i.e. Cs vs Sr in the same solution) have not been conducted.

[R3, Response #9]

It is reasonable to reconsider the usage of the word "selective" since our added competitive experiments results reveal that ion-exchange capacity drops at Na⁺ rich environment. Thus we replaced to more appropriate wording when describing the functionality of NMSCs.

✓ **Changes made corresponding to the word "selective"**

“In addition, these chalcogels successfully adopted the structural motifs and ion-exchange principles of two-dimensional layered metal sulfides ($K_{2x}Mn_xSn_{3-x}S_6$, KMS-1), featuring a layer-by-layer stacking structure and **effective** radionuclide (Cs^+ , Sr^{2+})-control functionality.” (L31)

“To explore the potential functionality of NMSCs for **effective** adsorption, an aqueous Cs^+ remediation test was conducted.” (L444-445)

“Without further functionalizations, the soft basic surface and Na-anchoring sites of NMSCs is anticipated to bound intermediate-to-soft cations of Cs^+ and Sr^{2+} as other precedent chalcogenide aerogels have done successfully¹³.” (L453-455)

“This new class of aerogels possesses the same microstructure as previously reported for layered metal sulfides (KMS-1), which are known for their extraordinary radionuclide (Cs^+)-remediation functionality via the hard–soft-acid–base (HSAB) ion-exchange principle. Sharing the common structural features of the anionic Mn-Sn-S bridge layer charge-balanced by alkali metal ions, NMSCs inherit its ion-exchange capabilities which can be used for **a selective and** an effective radionuclide control.” (L567-572)

[R3, Comment #10]

Beginning on line 421 the authors state “The seemingly unmatched sodium–sulfide pair, counterintuitively, exhibited a good ion-exchange functionality, rendering Na^+ a good leaving group upon dissolution in the aqueous medium.” in reference to the performance of these materials in exchanging Na^+ ions for Cs^+ ions. Describing this observation as counterintuitive is confusing as the “unmatched sodium-sulfide pair” is the reason why this material is able to exchange in the first place, which would make the performance intuitive. I ask the authors to clarify the wording of this sentence.

[R3, Response #10]

Thank you for the feedback. We also agree with the author’s point that our previous statement regarding the ion-exchange functionality was too ambiguous and confusing. We rewrote the sentence with understandable wording for lesser confusion as following:

✓ NMSCs ion-exchange potential description (L451-455)

“In contrast, NMSCs are anticipated to be functional for Cs⁺ and Sr²⁺ capture through ion-exchange principle as precedent layered chalcogenide compound (e.g., KMS-1) are known to attract them with high efficiency. Without further functionalizations, the soft basic surface and Na-anchoring sites of NMSCs is anticipated to bound intermediate-to-soft cations of Cs⁺ and Sr²⁺ as other precedent chalcogenide aerogels have done successfully¹³.”

[R3, Comment #11]

Beginning on line 465, the authors state “At pH 2–10, NMSC-1 exhibited 37.4–56.1% of the original Cs⁺-removal capacity (Fig. 5d), indicating a potential ion-exchange competition with H₃O⁺ under acidic conditions and hydroxyl group attacks on the sulfidic sites under basic conditions. This is a unique feature of NMSCs, which was not observed in the original layered metal sulfide (KMS-1) structural motif.” This statement is untrue as in the original report of KMS-1 (ref 27 in the manuscript), the variable pH Cs ion exchange study conducted demonstrated removal percentages of ~42-76% of the theoretical capacity over the pH range of 1-12. Furthermore, it is imprecise to describe these results by saying “At pH 2–10, NMSC-1 exhibited 37.4–56.1% of the original Cs⁺ removal capacity” as the this experiment does not determine the capacity of the material at different pH’s, these are more precisely removal percentages. To make this statement one would need to conduct isotherms at each pH of interest to comment on the change in capacity over pH. I suggest the authors rephrase their statement to something like “...exhibited removal percentages of 37.4–56.1% in reference to the capacity determined at pH=7”.

[R3, Response #11]

Thank you for the detailed and critical feedback regarding the evaluation of NMSC's ion-exchange functionality. We admit that ion-exchange competition phenomena on NMSC's could be universal feature of layered chalcogenide structure. Also, the description of Cs⁺ removal at various pH condition was not clear enough. So we adjusted the following areas:

- ✓ Corrections related to the pH dependent radionuclide removal

~~“This is a unique feature of NMSCs, which was not observed in the original layered metal sulfide (KMS-1) structural motif.”~~

~~“At pH 2–10, NMSC-1 exhibited 37.4–56.1% (\pm 4-26 mg g⁻¹ standard deviation) of the original Cs⁺-removal capacity (Fig. 5d), ” (L559-560)~~

REVIEWERS' COMMENTS

Reviewer #1 (Remarks to the Author):

The reviewer is satisfied with the changes made by the authors in the revised manuscript and is willing to recommend the manuscript by Kim et al. to be published in Nature Communications. Some minor issues should be corrected.

1. Supplementary Figure 19, (c-d) NMgSC-3 ($\text{Mg}^{2+}:\text{Sn}_2\text{S}_6 = 2:1$)
2. Ref. 24, 2022 1, 220046.

Reviewer #2 (Remarks to the Author):

The authors have well responded to my concerns.

Reviewer #3 (Remarks to the Author):

The authors made an excellent job in responding to my questions and concerns. I am satisfied with the revisions and I recommend acceptance without any changes. It is now a great paper.

We thank the reviewers for their helpful advices, which strengthen the scientific importance of our work. In the following, we addressed point by point all the comments made by the reviewers to our manuscript (NCOMMS-22-28388A). All revisions to the manuscripts addressing the comments of the Reviewers are highlighted in the blue text.

Response to the Reviewer 1's comment in the second review

[Comment]

The reviewer is satisfied with the changes made by the authors in the revised manuscript and is willing to recommend the manuscript by Kim et al. to be published in Nature Communications.

Some minor issues should be corrected.

[R1, Comments #1]

Supplementary Figure 19, (c-d) NMgSC-3 ($\text{Mg}^{2+}:\text{Sn}_2\text{S}_6 = 2:1$)

[R1, Response #1]

We thank for constructive comment to clarify scientific importance of our work. We revised the label of NMgSC-3 in Supplementary Figure 19. Changes are listed as follows:

- ✓ Revised Supplementary Information (P 24)

Supplementary Figure 19. Morphological analysis of NMgSC and NSn(II)SC aerogel. FE-SEM results of **a-b** NMgSC-1 ($\text{Mg}^{2+}:\text{Sn}_2\text{S}_6 = 0.5:1$), **c-d** NMgSC-2 ($\text{Mg}^{2+}:\text{Sn}_2\text{S}_6 = 2:1$), **e-f** NSn(II)SC-1 ($\text{Sn}^{2+}:\text{Sn}_2\text{S}_6 = 0.25:1$) and **g-h** NSn(II)SC-2 ($\text{Sn}^{2+}:\text{Sn}_2\text{S}_6 = 2:1$).

[R1, Comment #2]

Ref. 24, 2022 1, 220046.

[R1, Response #2]

Thank you for your kind comment. According to your suggestion, we revised the Ref. 24 in the main manuscript. The changes are listed as follows:

- ✓ Revised References in main manuscript (L 737-738)
24. Bai J., Yang L., Zhang Y., Sun X., Liu J. Tin Sulfide Chalcogel Derived SnS_x for CO₂ Electroreduction. *Materials Lab* **1**, 220046 (2022).

Authors revised the errors in typos in the manuscript

There are typos in the main manuscript. Thus, we revised them to more accurate wording. The changes are listed as following:

- ✓ Revised main manuscript
- "...[Sn₂S₆]⁴⁻ turned to a light-amber turbid solution over 5 h, and subsequently to a darker wet..." (L 121)
- "The typical metal layer sulfides (KMSn₂S₆) families in the space group of $R\bar{3}m$ includes" (L 249)
- "...manganese-rich environment prevents the assembly of the tetrahedral MTC units..." (L 386)
- "...paramagnetic behavior than the mixed tetrahedral-octahedral coordination macrostructure (NMSC-3)." (L 408-409)
- "...formamide at room temperature (25 °C)." (L 592)
- "For Cs⁺ and Sr²⁺ removal experiments, 100 ppm of 100 mL Cs⁺ or Sr²⁺ aqueous solutions wer prepared using CsCl or SrCl₂, respectively. After stirring of 15 mg of NMSCs in 100 mL, 100 ppm CsCl or SrCl₂ aqueous solution for 3 h at 25 °C, all solutions were filtered for Cs⁺ or Sr²⁺ concentration detection." (L 629-632)
- "Solid-state ¹¹⁹Sn NMR analysis" (L 638)
- "MEMS·Sensor Platform Center of Sungkyunkwan University." (L 659)

- ✓ Revised Supplementary Information

There are typos in the main manuscript. Thus, we revised them to more accurate wording. The changes are listed as following:

- “The 1 g of dried NMgSC-1 was grinded and pelletized under the pressure of 350 bar to make the flatten surface of sample” (P 4)
- “The Na⁺-Cs⁺ and Na⁺-Sr²⁺ dual ion solutions are prepared at two different concentration ratios. The Cs⁺ initial concentration of solutions is 10 ppm. The Na⁺ initial concentration is variable at 10 ppm and 100 ppm in Na⁺-Cs⁺ dual ion solution batches (respectively, Na⁺:Cs⁺= 1:1 and Na⁺:Cs⁺= 10:1). The Sr²⁺ initial concentration of solutions is 10 ppm. The Na⁺ initial concentration is variable at 10 ppm and 100 ppm in Na⁺-Sr²⁺ dual ion solution batches (respectively, Na⁺:Sr²⁺= 1:1 and Na⁺:Sr²⁺= 10:1).” (P 4)
- Authors revised the label of NMgSC and NSn(II)SC in Supplementary Figure 18. (P 23)

Supplementary Figure 18. Chemical bonding-nature of NMgSC and NSn(II)SC samples. Raman spectroscopy analysis of NMgSC and NSn(II)SC aerogels at different [Sn₂S₆]⁴⁻:Mg²⁺/Sn²⁺ ratios. Peaks at 257, 339 and 356 cm⁻¹ correspond to Sn₂S₂ ring and Sn-S octahedral vibrations at 532 nm.